# From comprehensive study to low rank compensation: Exploring post-training quantization in LLMs

## Abstract

Post-training quantization (PTQ) has emerged as a promising technique for mitigating memory consumption and computational costs in large language models (LLMs). However, a systematic examination of various quantization schemes, model families, and quantization bit precision has been absent from the literature. In this paper, we conduct a comprehensive analysis of these factors by investigating the effects of PTQ on weight-only, activation-only, and weight-and-activation quantization using diverse methods such as round-to-nearest (RTN), GPTQ, ZeroQuant, and their variants. We apply these methods to two distinct model families with parameters ranging from 125M to 176B. Our contributions include: (1) a sensitivity analysis revealing that activation quantization is generally more susceptible to weight quantization, with smaller models often outperforming larger models in terms of activation quantization; (2) an evaluation and comparison of existing PTQ methods to optimize model size reduction while minimizing the impact on accuracy, revealing that none of the current methods can achieve the original model quality for quantization with either INT4-weight or INT4-weight-and-INT8-activation; (3) based on these insights, we propose an optimized method called Low-Rank Compensation (LoRC), which employs low-rank matrices to enhance model quality recovery with a minimal increase in model size.

## 1 Introduction

Large language models (LLMs) like Codex [15] and ChatGPT [24] have demonstrated breakthrough performance across various benchmarks, such as natural language understanding and generation, and are now integrated into everyday applications. However, efficiently serving LLMs has become a pressing concern due to their significant memory consumption and computational demands. Unlike classification or diffusion models, LLMs present unique challenges, as they involve two distinct phases: prompt and generation. The prompt phase is primarily compute-bound, while the generation phase, with low batch size and KV cache, is mainly memory-bound [26].

As the progression of hardware bandwidth lags behind that of computational demand [14], the resource demands of extra-large models such as MT-NLG-530B [30]—which necessitates the deployment of multiple nodes for operation—escalate, adding to the complexities of cross-node communication. This has emphasized the urgency to curtail both the size and computational expense of Large Language Models (LLMs). An increasingly effective solution to these issues is post-training quantization (PTQ). This method aids in the reduction of training prerequisites while simultaneously lowering the bit precision of weights and activations to either INT4 or INT8.

While the effectiveness of post-training quantization (PTQ) has been underscored in a number of recent studies [36, 12, 35, 7], a comprehensive, systematic investigation into several key dimensions of this technique remains to be undertaken. Specifically, the extant literature falls short in providing

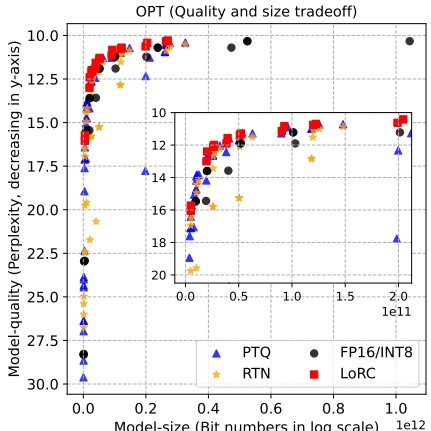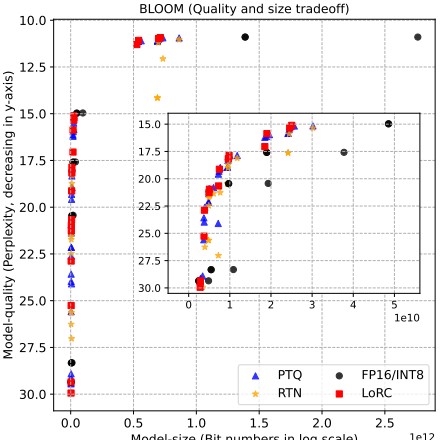

Figure 1: The model size and quality trade-off of different quantization methods on models from OPT and BLOOM families. Here PTQ (with fine-grained quantization) represents the method from [36, 12], RTN means the naive round-to-nearest baseline (with fine-grained quantization as well), and FP16/INT8 is used as the no-accuracy-loss baseline. LoRC is our proposed method that works seamless with PTQ. Note that we drop all diverged points for better visualization. For all detailed numbers, please see Appendix E.

thorough coverage of the functionality of various PTQ methods or the sensitivity of disparate models. Moreover, despite current quantization methods demonstrating promising results in the reduction of model sizes, the question persists as to whether these methods are achieving their optimal potential in minimizing Large Language Models (LLMs) sizes.

With these observations in mind, our study sets forth to address two salient questions: (1) When subjected to quantization, do LLMs of varying sizes and pretraining data exhibit similar behavior? (2) Are existing quantization methods truly leveraging their full potential in reducing the sizes of LLMs?

**Contribution.** To elucidate these queries, we undertake an exhaustive examination of the impact of PTQ on weight-only, activation-only, and combined weight-and-activation quantization. This investigation incorporates a range of PTQ methods, including round-to-nearest (RTN), GPTQ [12], ZeroQuant [36], and their respective variants. To broaden the scope of our analysis, we focus on two distinct model families, OPT [40] and BLOOM [28], spanning model sizes from 125M to a massive 176B. Our code will be made available for reproduction. In summary, we make the following contributions:

(1) We provide a thorough **sensitivity analysis** to demonstrate that a) Activation quantization is generally more sensitive to weight quantization; Smaller models usually have better activation quantization performance than the relative larger model. b) Different model families show different INT8 activation quantization behaviors; Particularly for large models, BLOOM-176B has small accuracy drops (about 1 perplexity or PPL) but OPT-30B and -66B experience worse performance.

(2) We carry out a detailed evaluation and comparison of current PTQ methods, utilizing optimal configurations to maximize model size reduction while minimizing accuracy impact. We found that the current existing method can barely achieve less than 0.1 PPL points degradation for quantization with either INT4-weight or INT4-weight-and-INT8-activation (W4A8). To recover the 0.1 PPL, we strive to push the boundaries of employing **fine-grained quantization** (FGQ) techniques. We observe FGQ is able to recovered points degradation of <0.1 PPL for large models (>13B) for INT4 weight quantization, but there are still non-negligible model quality drops.

(3) Based on the above understanding, we further optimize existing methods and introduce a technique called **Lo**w **R**ank **C**ompensation (LoRC), which employs low-rank matrix factorization on the quantization error matrix. Complementary to FGQ, LoRC plays a crucial role in enhancing the full model quality recovery, while there is little increase of the model size.

In Figure 1, we provide model size and quality trade-offs for both OPT and BLOOM families. As can be seen, using LoRC on top of PTQ methods from [36, 12] and fine-grained quantization,

we set a new quantization Pareto frontier for LLMs. Meanwhile, we recommend the following setting for quantizing LLMs with LoRC (Note that activation quantization should be only applied if necessary): (1) For larger models (>10B), fine-grained (block size 64–256) 4-bit weight quantization plus 8-bit activation quantization (block size 64–256) with PTQ can be used for real deployment; (2) For middle-size models (<10B and >1B), per-row INT8 quantization plus fine-grained (block size 64–256) INT8 activation quantization can be used with PTQ from [12, 36]; (3) For smaller models (<1B), per-row W8A8 (INT8 weight and INT8 activation) RTN is enough based on [36].

## 2 Related Work

Different quantization methods [29, 38, 9, 41, 1, 8, 31, 19] for transformer-based models [32] have been explored for a while. However, most of those works need quantization-aware finetuning or even expensive quantization-aware knowledge distillation [17]. Due to the cost of training/finetuning LLMs [25, 18, 31, 34, 33], it is a challenge for practitioners/researchers to do finetuning/distillation on those LLMs, particularly for models like GPT-3-175B [4] and BLOOM-176B [28].

Post-training quantization (PTQ) [37, 3] is an alternative way to quantize the model with no/minimal finetuning requirement. Along this line, several recent works focus on LLMs (beyond the million-parameter scale). [36] proposes vector-based INT8 quantization with layer-by-layer knowledge distillation to overcome the training cost and quantization error introduced by LLMs. [6] uses similar vector-based INT8 quantization weight plus mixed-precision (INT8/FP16) quantization for activation to overcome the sensitivity of activation quantization. However, the inference speed of [6] is generally even slower than FP16 baseline [2] due to the difficulty of implementing mixed-precision calculation within a single tensor. More recently, [12] extends OBQ [10, 16, 21] on LLMs for INT4 weight-only quantization and shows great efficiency on quantization and latency, and [35] shows the outliers from activations can be smoothed out by migrating the quantization difficulty from activations to its associated weights. However, [35] can only work for W8A8 quantization as lower weight precision (INT4) itself already leads to significant accuracy degradation, and the accuracy drop is larger than 0.1 PPL points, which as discussed in the later section is sub-optimal. [7] shows the scaling law of weight-only quantization with the simplest round-to-nearest baseline, but it does not consider the weight-and-activation quantization and/or the above PTQ optimization methods. As can be seen from Figure 1, by using PTQ optimization methods, the model quality can be significantly improved. Please also see Appendix E for more detailed numbers.

Different than existing works, our paper extensively tests the effect of (1) different quantization schemes, e.g., symmetric and asymmetric quantization, (2) different PTQ methods, e.g., [36, 12], (3) different model families, e.g., [28, 40], (4) different quantization coverage, e.g., weight-only and weight-and-activation quantization, and (5) other discussions, e.g., the effect of quantization granularity. As such, we provide a much more comprehensive understanding of post-training quantization for large language models compared to the previous works.

## 3 Would different model families behave similarly on quantization?

There are mainly two categories of PTQ for LLMs, i.e., weight-only quantization [12] and weight-and-activation quantization [6, 36, 35]. In the latter, it is uniformly observed across all studies that activation quantization demonstrates greater sensitivity than weight quantization. However, prior research tends to concentrate on a single (family) model to emphasize the necessity of their proposed quantization technique. A comprehensive and systematic evaluation of this PTQ methodology, particularly the sensitivity of weight/activation quantization for varying model sizes and distinct model families, has yet to be undertaken. Hence, we conduct an examination on both the OPT [40] and BLOOM [28] families to elucidate the quantization sensitivity of weight and activation.

**Sensitivity setting.** We use the zero-shot validation perplexity (PPL) differential on three datasets, namely, Wikitext-2 [23], PTB [22], and C4 [27], before and after the quantization of these LLMs to illustrate their sensitivity, as PPL is significantly correlated to zero-shot/few-shot accuracy measurement [7]. Specifically, a higher PPL drop indicates enhanced quantization sen-

Table 1: Classification of quantization sensitivity (or quantization loss). The sensitivity increases from *Class*-1 to *Class*-3.

| Class | *Class*-1 | *Class*-2 | *Class*-3 |
|---|---|---|---|
| PPL Degradation | ≤0.1 | >0.1 & ≤0.5 | >0.5 |

sitivity. For simplicity, we also categorize quantization sensitivity (or quantization loss) into three different classes as depicted in Table 1. Notably, the threshold is chosen because when the model size approximately doubles (e.g., 13B vs. 30B, and 30B vs. 66B), the PPL improvement is about 0.5 (see Table 2). The sensitivity (or loss) incrementally increases as the class number ascends. From a practical standpoint, we favor lower quantization sensitivity (accuracy loss), making *Class*-1 the optimal-loss post-training quantization.

We employ both symmetric and asymmetric quantization to gauge the quantization sensitivity and highlight the advantage of asymmetric quantization. Particularly, we implement per-row quantization [12] for weight quantization and per-token quantization for activation [36].

**Robustness of Weight-only Quantization for Large Models.** The results of weight-only quantization in OPT and BLOOM models are summarized in Table 2. INT8 weight-only quantization, either symmetric or asymmetric, results in negligible accuracy loss (less than 0.05, i.e., *Class*-1). Consequently, for tasks oriented towards generation, FP16 weight can simply be replaced with INT8 weight to reduce memory usage. For INT4 quantization, the asymmetric method outperforms the symmetric approach in accuracy, attributable to its superior utilization of the quantization range. Interestingly, larger models exhibit better tolerance to low-precision quantization (i.e., INT4) than smaller models, with a few exceptions such as OPT-66B.[1] Particularly, BLOOM-176B shows PPL degradation (around 0.3 points) in *Class*-2, which could explain why the large GLM-130B [39] can operate with INT4 weight-only quantization out of the box with acceptable accuracy impact.

Table 2: Average PPL of OPT and BLOOM (BLM). See Table E.1 for all results.

| Precision | OPT-6.7b | OPT-13b | OPT-30b | OPT-66b | BLM-1.7b | BLM-3b | BLM-7.1b | BLM-176b |
|---|---|---|---|---|---|---|---|---|
| W16-A16 | 11.90 | 11.22 | 10.70 | 10.33 | 20.43 | 17.58 | 14.96 | 10.90 |
| W8$^{sym}$-A16 | 11.90 | 11.22 | 10.70 | 10.33 | 20.43 | 17.59 | 14.97 | 10.90 |
| W8$^{asym}$-A16 | 11.90 | 11.22 | 10.70 | 10.33 | 20.45 | 17.59 | 14.97 | 10.90 |
| W4$^{sym}$-A16 | 14.36 | 12.73 | 11.77 | 97.05 | 23.18 | 19.36 | 16.27 | 11.28 |
| W4$^{asym}$-A16 | 13.44 | 12.09 | 11.52 | 31.52 | 22.47 | 19.01 | 15.90 | 11.20 |
| W16-A8$^{sym}$ | 26.04 | 3171.49 | 2048.21 | 2638.09 | 20.68 | 17.73 | 15.28 | 12.10 |
| W16-A8$^{asym}$ | 12.62 | 15.36 | 23.57 | 561.35 | 20.52 | 17.65 | 15.14 | 11.62 |

**Challenge Encountered in Activation Quantization for Large Models.** Activation quantization has consistently proven more difficult than weight quantization [36, 6], as illustrated in Table 2. When compared to weight-only quantization, activation-only quantization indicates that asymmetric quantization can significantly improved performance over symmetric quantization. Moreover, contrary to weight-only quantization, smaller models typically exhibit better tolerance to activation quantization, as their hidden dimension is smaller and the activation dynamic range is also narrower than larger models [36]. It should be noted that for models larger than 10B, all fall into *Class*-3, indicating a degradation of more than 0.5 PPL points.

The last two rows of Table 2 show that different model families exhibit significantly different behaviors. BLOOM does not exhibit divergence issues even up to a model size of 176B, whereas OPT displays very poor performance from a model size of 6.7B (larger models with INT8 activation have even worse PPL). This could again be attributed to the Layer Norm issue within the OPT-family[1].

> **Findings 1 on Sensitivity Analysis.** (**1**) INT8 weight-only quantization can serve as a standard method for reducing memory costs in LLMs, with negligible degradation in accuracy. (**2**) INT4 weight-only quantization for small models results in substantial accuracy degradation (*Class*-3), but this effect lessens as the model size increases (*Class*-2). (**3**) Contrary to (2), INT8 activation results in minimal accuracy drops for small models (*Class*-1) but larger models exhibit greater drops (*Class*-3). (**4**) With INT8 activation, BLOOM shows no divergence issues up to a model size of 176B, whereas OPT performs poorly from $\geq$ 6.7B model sizes.

---

[1][12] discovered that OPT-66B has a high proportion of dead neurons in the early layers, which might influence the compression capability. We also identify another potential reason: the Layer Norm of the OPT-family is not well trained (except OPT-350M), with the weight and the bias being all 1's and 0's, respectively.

## 4 Are existing quantization methods optimally harnessing the potential to minimize LLMs sizes?

Numerous lightweight optimization-based methods have been proposed, which update the model weights during quantization. These methods such as [36, 12, 35], unlike quantization-aware training, only require a small portion of the training data and a limited training time. Particularly, GPTQ [12] and ZeroQuant [36], have proven to be effective and efficient in terms of GPU resources, time cost, and data usage for INT4 weight quantization.[2] In this work, we focus on the variants of GPTQ and ZeroQuant as well as the most straightforward baseline, round-to-nearest neighborhood (RTN).

**RTN** directly applies PTQ on the trained data and follows the procedure detailed in Section A to perform the quantization. Specifically, for symmetric quantization, we set $S = max(abs(x))$ and $Z = 0$; for asymmetric quantization, we set $S = max(x) - min(x)$ and $Z = min(x)$.

**GPTQ** extends the OBQ [10]. It tries to optimize the following non-linear least square problem, $\min_{\hat{W}} \|Wx - \hat{W}x\|_2^2$ where $W$ is the weight, $x$ is the activation, and $\hat{W}$ is a quantized weight. GPTQ employs second-order methods to obtain a closed-form solution. In addition, the quantization for each weight matrix is performed column-/row-wisely and the quantization errors from previous columns will be passed to those columns not yet quantized. See[10, 12] for more details.

**ZQ-Global** is the original method proposed in [36], where authors treat each layer as a small neural network (a.k.a., subnetwork) and use the FP16 subnetwork as the teacher model to distill the quantized one with a few hundred iterations, i.e., $\min_{\hat{\theta}} |f_\theta(x) - f_{\hat{\theta}}(x)|2^2$, where $\theta$ is a set of weights, $\hat{\theta}$ is the quantized version, $f\theta$ is the subnetwork with parameters $\theta$, and $x$ is the input. Thus, it can significantly reduce the GPU resource requirement and time cost.

**ZQ-Local** is an extension mode of ZQ-Global for further GPU requirement reduction and training cost reduction. Particularly, instead of using each transformer layer as the subnetwork, we treat each linear layer as the subnetwork. This method can be viewed as an iterative first-order optimization method (e.g., SGD) to solve $\min_{\hat{W}} \|Wx - \hat{W}x\|_2^2$.

**Experimental Setup.** We compare the four methods mentioned above on weight-only and weight-and-activation quantization. As weight quantization is always static (i.e., it does not change during inference), there is virtually no system performance difference between symmetric and asymmetric quantization.[3] We use asymmetric quantization for better accuracy, and the conclusions would hold similarly for symmetric quantization. For parameters used for GPTQ, ZQ-Local, and ZQ-Global, please refer to Appendix B. An interesting finding for ZeroQuant is that the hyperparameters (e.g., learning rate and its scheduler) provided in the original work [36] are sub-optimal. In this work, we find the best configurations for ZQ-Local and ZQ-Global and denote them as ZQ-Local* and ZQ-Global*, respectively, with the best tuned results. To ensure consistent and comparable results, we set a fixed random seed for our experiments. In the context of post-training quantization, varying the random seed has minimal impact on the final results, as indicated in more detail in Table B.1.

**Evaluation of Weight-only Quantization.** The results from weight-only quantization using OPT and Bloom are presented in Table 3. The findings indicate that the larger models tend to be less sensitive to INT4 weight-only quantization. This observation holds true across all methods (RTN, GPTQ, ZQ-Local*, and ZQ-Global*) with the exception of OPT-66B, which shows greater degradation than OPT-30B. It is noteworthy that light-weight optimization-based methods significantly outperform the RTN baseline in terms of accuracy. For instance, these methods substantially reduce the degradation in perplexity of OPT-30B/66B compared to baseline. Most quantized models with parameters greater than 6.7B fall under Class II, indicating their potential for real-world applications. For instance, the quality of INT4 OPT-30B (66B) is superior to that of INT8 OPT-13B (30B).

Among the optimization-based methods, ZQ-Global* generally performs better on smaller models (those with fewer than 1B parameters), while GPTQ excels on larger models. ZQ-Local* does not outperform GPTQ or ZQ-Global*-— a reasonable outcome given that GPTQ employs a closed-form solution to solve the non-linear quadratic problem and ZQ-Global* optimizes a larger subnetwork. The inferior performance of ZQ-Global* compared to GPTQ for larger models is unexpected since ZQ-Global* optimizes an entire transformer layer while GPTQ only optimizes a single linear layer.

---

[2]We tested the method proposed by [35] but did not find it better than others for INT4 weight quantization.
[3]The bias term (a.k.a., the zero point) can be simply fused into the previous activation quantization kernel [36].

Table 3: The evaluation results of different PTQ methods on OPT and BLOOM (BLM) with asymmmetric quantization on weight or (and) activation. See more details in  Table E.3 and Table E.6.

| Precision | Method | OPT-6.7b | OPT-13b | OPT-30b | OPT-66b | BLM-1.7b | BLM-3b | BLM-7.1b | BLM-176b |
|-----------|--------|----------|---------|---------|---------|----------|--------|----------|----------|
| W16A16 | | 11.90 | 11.22 | 10.70 | 10.33 | 20.43 | 17.58 | 14.96 | 10.90 |
| W4A16 | RTN | 13.44 | 12.09 | 11.52 | 31.52 | 22.47 | 19.01 | 15.90 | 11.20 |
| | GPTQ | 12.28 | 11.42 | 10.78 | 10.52 | 21.58 | 18.33 | 15.50 | 11.02 |
| | ZQ-Local* | 12.46 | 11.64 | 11.05 | 10.79 | 21.70 | 18.50 | 15.55 | 11.11 |
| | ZQ-Global* | 12.38 | 11.62 | 11.04 | 10.68 | 21.38 | 18.33 | 15.52 | 11.05 |
| W4A8 | RTN | 14.80 | 26.36 | 86.26 | 815.00 | 22.75 | 19.17 | 16.19 | 12.22 |
| | GPTQ | 13.88 | 17.28 | 20.71 | 648.69 | 21.71 | 18.44 | 15.75 | 11.86 |
| | ZQ-Local* | 13.24 | 14.23 | 18.53 | 16.32 | 21.86 | 18.66 | 15.75 | 11.19 |
| | ZQ-Global* | 13.17 | 13.07 | 14.65 | 37.82 | 21.43 | 18.39 | 15.58 | 11.49 |

A plausible explanation is that larger models are more sensitive to weight updates, necessitating more advanced fine-tuning methods.

**Evaluation of Weight and Activation Quantization.** The evaluation results for existing methods using W4A8 quantization are presented in Table 3.  The three light-weight optimization-based methods outperform RTN significantly, underscoring their efficacy. However, all of the results fall into either *Class*-2 or *Class*-3. This suggests that for certain applications, it might be more beneficial to use smaller models with fewer parameters rather than larger, quantized models.

Among quantization-based methods, ZQ-Global* and ZQ-Local* generally outperform GPTQ, which is anticipated given that GPTQ was originally designed for weight-only quantization. ZQ-Global* performs better than ZQ-Local* in most cases except for the two largest models, OPT-66B and Bloom-176B, despite having larger trainable parameters in one step. This again signifies the need for a more suitable and advanced optimization method for large language models (LLMs).

> **Finding 2 on Comparisons. (1)** GPTQ typically performs better for weight-only quantization, while ZeroQuant (including both ZQ-Global* and ZQ-Local*) yields superior results for weight and activation quantization. **(2)** The tested optimization-based methods cannot achieve *Class*-1 quantization error for either INT4 weight-only or W4A8 quantization with the exception of GPTQ on OPT-30B with weight-only quantization.

## 4.1   Fine-grained Quantization and Its Evaluation

With PTQ and row-wise quantization, achieving *Class*-1 quantization error is challenging for both weight-only and weight-and-activation quantization. Generally, utilizing a smaller model with INT8 weight is more advantageous than employing a model that is twice as large with INT4 weight.

One potential solution to this issue is the implementation of finer-grained quantization schemes [5], where every $k$ elements possess their own scaling factor and/or zero point.  This approach can significantly reduce quantization error. In the extreme case, where every single element has its own scaling factor, the original FP16 number can be precisely recovered. Importantly, block-k quantization can be implemented on modern GPUs, one of the most prevalent deep learning architectures, since the compute unit (streaming multiprocessor) of GPUs processes tiles of data (e.g., 128 by 128 tiling size) for matrix computation.

Although fine-grained quantization can substantially narrow the gap between the quantized tensor and its floating-point counterpart, the application of RTN still results in a non-trivial accuracy gap. Consequently, we build upon fine-grained quantization by employing existing optimization-based methods to further enhance accuracy. Specifically, we utilize GPTQ and ZQ-Global for all models and settings and apply ZQ-Local to OPT-66B and Bloom-176B. For the hyperparameters used in ZQ-Global and ZQ-Local, we select the top three identified in Section 4 for all models, except for Bloom-176B, for which we only use the top-performing hyperparameter to reduce training costs.

**4-bit Weight Quantization.**   We hereby present the W4A16 results for OPT and BLOOM, as delineated in Table 4, corresponding to an array of quantization block sizes. The performance sees a significant improvement with smaller block sizes compared to per-row quantization. The point of diminishing returns, however, varies for different model sizes. For example, smaller models (such as OPT-6.7B and BLOOM-1.7b) continue to see substantial gains until the block size reduces to 32. In contrast, for larger models (those exceeding 10B, with OPT-66B as the excep-

Table 4: Results of **W4asym-A16** quantization with various block-size out of the best result from optimization-based methods on OPT and BLOOM (BLM). See Table E.15 and Table E.16 for full results including RTN. N/A means that the block size is not divisible by the hidden size.

| Block-size | OPT-6.7b | OPT-13b | OPT-30b | OPT-66b | BLM-1.7b | BLM-3b | BLM-7.1b | BLM-176b |
|---|---|---|---|---|---|---|---|---|
| W16A16 | 11.90 | 11.22 | 10.70 | 10.33 | 20.43 | 17.58 | 14.96 | 10.90 |
| Per-row | 12.28 | 11.42 | 10.78 | 10.52 | 21.38 | 18.33 | 15.50 | 11.02 |
| 1024 | 12.16 | 11.36 | 10.75 | 10.52 | 31.03 | N/A | 15.24 | 10.96 |
| 512 | 12.08 | 11.32 | 10.73 | 10.52 | 20.93 | 17.99 | 15.20 | 10.95 |
| 256 | 12.05 | 11.28 | 10.74 | 10.50 | 20.95 | 17.97 | 15.18 | 10.95 |
| 128 | 12.10 | 11.28 | 10.74 | 10.44 | 20.92 | 17.90 | 15.17 | 10.94 |
| 32 | 12.03 | 11.28 | 10.72 | 10.41 | 20.82 | 17.88 | 15.16 | 10.95 |

Table 5: OPT W4asym-A8 with various block-size out of the best result from GPTQ, ZQ-Local, and ZQ-Global on OPT and BLOOM (BLM). See Table E.20 for full results including RTN.

| Precision | block-size (W\|A) | OPT-6.7b | OPT-13b | OPT-30b | OPT-66b | BLM-1.7b | BLM-3b | BLM-7.1b | BLM-176b |
|---|---|---|---|---|---|---|---|---|---|
| W4A16 | 128 \| NA | 12.10 | 11.28 | 10.74 | 10.44 | 20.92 | 17.90 | 15.17 | 10.94 |
| W4A8 | Case-1: per-row \| per-row | 13.17 | 13.07 | 14.65 | 16.32 | 21.43 | 18.39 | 15.58 | 11.19 |
| | Case-2: per-row \| 128 | 12.29 | 11.45 | 10.80 | 10.61 | 21.59 | 18.31 | 15.52 | 11.03 |
| | Case-3: 128 \| 128 | 12.04 | 11.31 | 10.75 | 10.45 | 21.27 | 17.86 | 15.19 | 10.96 |

tion), the benefits derived from smaller block sizes wane rapidly around block-256/512. Most crucially, for models equal to or larger than 13B, a smaller quantization block size results in quantization error being classified under *Class*-1, indicating virtually negligible degradation in accuracy.

**Activation Quantization (W4A8).** To comprehend the benefits of fine-grained quantization on activation, we analyze the quantization between per-row and a block size of 128, with INT4 weight, as highlighted in Table 5. For models of considerable size, specifically those equal to or exceeding 1B, the application of such fine-grained activation quantization (Case-1) results in a

Table 6: BLOOM-176B with different quantization block sizes on activation. Here weight is asymmetrically quantized with block size 128. See more in Table E.22.

| A8 Block Size | 1024 | 512 | 256 | 128 | 32 |
|---|---|---|---|---|---|
| PPL | 10.98 | 10.97 | 10.95 | 10.95 | 10.95 |

substantial reduction in quantization error compared to per-row activation (Case-2). By implementing fine-grained activation quantization with weight quantization (Case-3), we are able to almost restore the performance to the level of their W4A16 counterparts.

Furthermore, we detail the impacts of varying activation quantization block sizes in Table 6 on BLOOM-176B, with INT4 weight. A trend of superior accuracy is observed with smaller block sizes in contrast to larger ones. However, the enhancement in performance reaches a saturation point when the size smaller or equal to 256, which corresponds to the range of values INT8 can represent. Despite INT8's capability to signify 256 distinct values, activation quantization errors persist due to the application of uniform quantization.

> **Finding 3 on FGQ. (1)** Larger models ($\geq$10B) are capable of attaining *Class*-1 error for 4-bit quantization. These models can leverage low-precision quantization as the model size with INT4 is similar to an INT8 model that is half its size, with improved accuracy. On the other hand, smaller models ($\leq$10B) typically reach only *Class*-2 or *Class*-3 error levels. **(2)** For larger models (>10B), the difference between fine-grained weight-and-activation quantization and fine-grained weight-only quantization is insignificant. **(3)** The advantage of fine-grained activation quantization fades for larger models when the block size reaches 256.

## 5  Proposed Method to Further Push the Limit of Post-training Quantization

Building on the investigation and conclusions drawn from previous sections, it has become apparent that there is still a need for an advanced methodology to further refine the existing methods, with the objective of fully realizing the original fp16 PPL quality. In this section, we introduce a simple yet effective method called **LoRC** (Low Rank Compensation) to optimize the current existing quantization error and further bridge the gap between the quality of the original model and its quantized counterparts.

Table 7: W#$^{\text{asym}}$-A16 quantization with # being 4-bit, 3-bit and 2-bit on OPT and BLOOM (BLM).

| Bits | LoRC | Coarse-grained weight quantization (per-row block-size) | | | | | Fine-grained quantization on weight (256 block-size ) | | | | |
|---|---|---|---|---|---|---|---|---|---|---|---|
| | | OPT-6.7b | OPT-13b | OPT-30b | OPT-66b | BLM-176b | OPT-6.7b | OPT-13b | OPT-30b | OPT-66b | BLM-176b |
| W8A16 | | 11.90 | 11.22 | 10.70 | 10.33 | 10.90 | 11.90 | 11.22 | 10.70 | 10.33 | 10.90 |
| W4A16 | ✗ | 12.28 | 11.42 | 10.78 | 10.78 | 11.02 | 12.05 | 11.28 | 10.74 | 10.50 | 10.95 |
| W4A16 | ✓ | 12.10 | 11.36 | 10.76 | 10.34 | 10.98 | 11.99 | 11.29 | 10.70 | 10.29 | 10.93 |
| W3A16 | ✗ | 14.18 | 12.43 | 11.28 | 17.77 | 49.46 | 12.79 | 11.63 | 10.9 | 11.34 | 11.13 |
| W3A16 | ✓ | 13.00 | 11.90 | 11.14 | 10.63 | 11.30 | 12.40 | 11.57 | 10.83 | 10.42 | 11.08 |
| W2A16 | ✗ | 120.56 | 40.17 | 25.74 | 225.45 | Explode | 23.13 | 15.55 | 12.68 | 308.49 | 12.64 |
| W2A16 | ✓ | 24.17 | 18.53 | 14.39 | 13.01 | 14.15 | 16.27 | 14.30 | 12.37 | 11.54 | 12.21 |

LoRC is inspired by the employment of low-rank matrix factorization on the quantization error matrix $E := W - \hat{W}$, where $W$ represents the original weight and $\hat{W}$ is the quantized weight. LoRC approximates the error $E$ with $\hat{E} = \hat{U}\hat{V}$ by using two low-rank matrices $\hat{U}$ and $\hat{V}$. This results in a more accurate approximation of the original weight matrix $W$ by $\hat{W}_{\text{lorc}} = \hat{W} + \hat{E}$, thereby reducing quantization errors: $\|W - \hat{W}\| \geq \|W - \hat{W}_{\text{lorc}}\|$. LoRC consists of two steps:

**Step I:** Implement Singular Value Decomposition (SVD) on the error matrix $E = U\Sigma V$, where $U \in \mathbb{R}^{d_{\text{in}} \times d_{\text{in}}}$ and $V \in \mathbb{R}^{d_{\text{out}} \times d_{\text{out}}}$ are unitary matrices, and $\Sigma \in \mathbb{R}^{d_{\text{in}} \times d_{\text{out}}}$ is a diagonal matrix with its diagonal elements ordered in a descending manner.

**Step II:** We formulate the matrix $\hat{E} = \hat{U}\hat{V}$ where $\hat{U} = U_m(\Sigma_m)^{\frac{1}{2}}$ and $\hat{V} = (\Sigma_m)^{\frac{1}{2}}V_m$. Here, $U_m = U_{:,1:m} \in \mathbb{R}^{d_{\text{in}} \times m}$, $V_m = V_{1:m,:} \in \mathbb{R}^{m \times d_{\text{out}}}$, and $\Sigma_m = \Sigma_{1:m,1:m} \in \mathbb{R}^{m \times m}$.

The objective of LoRC is to achieve a good approximation of the error matrix $E$ using low-rank matrices, with minimal impact on the increase in model size. For instance, consider the standard transformer models [32], where each layer is comprised of a multi-headed attention (MHA) module and a multi-linear perception (MLP) module. Let $h$ represent the hidden dimension and $l$ the number of layers. The total number of parameters is $12lh^2$ as each layer contains $4h^2$ for MHA (for key, query, value, and projection matrices), and $8h^2$ for MLP (two matrices of sizes $h \times 4h$ and $4h \times h$). With the addition of low-rank LoRC to the six matrices in each layer, the total number of parameters for $l$ layers would amount to $18hml$.[4] Consequently, the ratio of parameters added to the existing model is $3m/2h$. It's important to note that the low-rank dimension $m$ can be as small as $4$ or $8$ (which we will discuss in detail in a later section) while the standard hidden dimension $h \geq 768$, making the number $3m/2h \leq 0.016$.

Significantly, LoRC can be viewed as a supplementary feature to existing quantization methodologies such as RTN, GPTQ, and ZeroQuant-local/Global, and can be seamlessly integrated with FGQ. We have conducted experiments to evaluate the performance of LoRC on both OPT and BLOOM, applying 4-bit, 3-bit, and 2-bit weights by setting the activation to FP16.[5] Based on the discoveries in the preceding sections, we utilize the GPTQ quantization strategy. To gain a comprehensive understanding of LoRC, we include the results with and without the application of FGQ. The datasets and hyperparameters are consistent with those detailed in earlier sections.

**Evaluation Results.** The findings are showcased in Table 7, split into two sections: coarse-grained weight quantization (per-row) and fine-grained quantization (block-size 256). Notably, we observe that the two low-rank matrices, $\hat{U}$ and $\hat{V}$, can be quantized to 8-bit without any performance discrepancy (Table 8). Thus, the two low-rank matrices for LoRC in Table 7 are INT8 with a low-rank dimension of $m = 8$.

Table 8: Results of W4$^{\text{asym}}$ A16 quantization with LoRC approximating $\hat{E} = \hat{U}\hat{V}$ on OPT model family. $\hat{U}$ and $\hat{V}$ can be represented with FP16 or INT8, of which the performance are represented below. There is hardly any difference between FP16 and INT8.

| LoRC $\hat{U},\hat{V}$ | Coarse-grained weight quantization | | | | Fain-grained weight Quantization | | |
|---|---|---|---|---|---|---|---|
| | 6.7b | 13b | 30b | 66b | 6.7b | 13b | 30b |
| FP16 | 12.08 | 11.35 | 10.76 | 10.31 | 11.993 | 11.290 | 10.703 |
| INT8 | 12.10 | 11.36 | 10.76 | 10.34 | 11.987 | 11.290 | 10.700 |

Several key observations can be made. Firstly, LoRC consistently boosts performance across all bit sizes and block sizes, as indicated by the lower perplexity scores when LoRC is activated. Secondly, the enhancement brought about by LoRC becomes more substantial as the bit size diminishes, especially noticeable for W2A16, which displays a markedly greater impact compared to W4A16 and W3A16 in most scenarios. Lastly, the

---

[4]In the MHA module, LoRC contributes $2hm$ to each of key, query, value, and the projection matrices. In the MLP module, LoRC contributes $8hm$ and $2hm$ respectively to the matrices of dimensions $h \times 4h$ and $4h \times h$.

[5]For INT8 Activation, please see Table E.23, the observation for FP16 holds similarly for INT8 Activation.

combination of fine-grained quantization with LoRC yields the most impressive results, underscoring the efficacy of LoRC when integrated with FGQ. Overall, the results emphasize the benefits of using LoRC for enhanced performance in weight quantization and its compatibility with FGQ. Notably, recovering the last 0.05-0.1 perplexity can be challenging, but with LoRC, we are able to nearly recover the original model quality for INT4 quantization.

Table 9: W4A16 quantization with LoRC by varying the low-rank dimension $m$.

| LoRC-dim $m$ | OPT-1.3b | OPT-6.7b | OPT-30b |
|---|---|---|---|
| $m = 0$ basline | 15.95 | 12.06 | 10.73 |
| $m = 1$ | 15.93 | 12.01 | 10.73 |
| $m = 4$ | 15.73 | 12.00 | 10.72 |
| $m = 8$ | 15.76 | 11.99 | 10.70 |
| $m = 16$ | 15.74 | 12.00 | 10.69 |
| $m = 32$ | 15.71 | 12.01 | 10.69 |

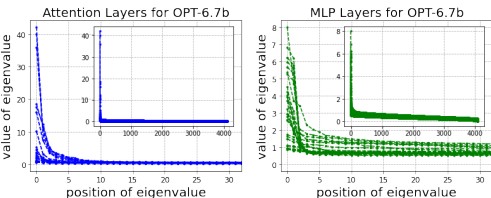

Figure 2: Eigenvalues of the Error matrix $E$ for W4A16

**Ablation Study on the Low Rank Dimension** $m$**.** An essential aspect of the LoRC method is on the optimal low-rank dimension, denoted as $m$, explained in **Step II**. To explore this, we varied $m$ in the range of 1, 4, 8, 16, and 32 for OPT-1.3b/6.7b/30b models, and applied W4A16 GPTQ quantization. The outcomes are depicted in Table 9, indicating that the enhancements achieved through LoRC begin to plateau as the dimension $m$ surpasses 4. The most optimal performance for OPT-6.7b is realized when $m = 8$.

This observation may seem counterintuitive initially, as one might anticipate that larger LoRC dimensions would yield more significant improvements. To gain a more comprehensive understanding, we conducted an analysis of the eigenvalues of the actual error matrix $E = W - \hat{W}$ for each matrix. By randomly selecting 20 matrices from MHA and MLP layers, we plotted the eigenvalues of $E$ as a curve, depicted in Figure 2. The two plots reveal a rapid flattening of eigenvalues after index 8, which elucidates why increasing the LoRC dimension does not considerably enhance performance. Hence, a sensible dimension for $\hat{U}$ and $\hat{V}$ in the LoRC methodology could be 8.[6]

## 6 Discussion

**Conclusion.** In this work, we provide a comprehensive study of post-training quantization (PTQ) on large language models with different PTQ methods (e.g., RTN, GPTQ, ZeroQuant), and with different quantization coverage (weight-only and weight-and-activation quantization), etc. We find that PTQ methods are critical to improving the quantized model quality, and that fine-grained quantization (FGQ) can bring acceptable accuracy and model size trade-off. Finally, we introduced an optimization technique called Low Rank Compensation (LoRC), which works synergistically with PTQ and FGQ, playing a crucial role in enhancing full model quality recovery with a minimal increase in model size.

**Limitation.** Despite quantizing over 10,000 experiments, our study was constrained by our computing resources. This restriction made us choose between diversifying the model sizes and varying the tasks. We strategically limited our datasets to WikiText, PTB, and C4 to concentrate on a broad range of quantization methods. Consequently, our general findings are more robust concerning the two model families and three datasets examined in this paper. However, caution should be exercised when generalizing these findings to tasks that are dissimilar to those covered in this study.

**Future Opportunity.** Throughout the paper, we see several unresolved problems from current quantization schemes and/or algorithms, and we find potential directions for LLM compression: (1) Although we use fine-grained quantization schemes in the paper, the real implementation is missing. How to efficiently implement odd bit precision is also challenging. [12] demonstrated that 3-bit can achieve better throughput in the generation phase by packing all 3-bit numbers in continuous memory space. However, this method is sub-optimal as the dequantization step needs to connect bits from different bytes. One possible way to implement odd bits, e.g., 5 bits, is to use two integer matrices with INT4 and INT1. During the dequantization stage, we couple the two matrices together. (2) How to combine PTQ with other lightweight compression techniques, e.g., post-training pruning [20, 11], is an interesting direction to further reduce the memory consumption and compute cost.

---

[6]Please note that this observation is only true for PTQ. If one uses quantize-aware training (QAT) and let $\hat{U}$ and $\hat{V}$ updated during QAT, we arrive at contrasting conclusions. For more details, please refer to Appendix D.

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
