## A  Background of Quantization

Quantization maps floating point (e.g., FP16/FP32) numbers to integer numbers (e.g., INT4/INT8) so that lower memory usage (weight quantization) and faster integer arithmetic (weight-and-activation quantization) can be achieved compared to the floating point format. In this work, we are focusing on uniform quantization, i.e.,

$$Q(x) = \text{INT}\big((x - Z)/S\big) - Z, \tag{1}$$

where $Q$ is the quantization function, $x$ is a floating point input vector/tensor, $S$ is a real valued scaling factor, and $Z$ is an integer zero point. Based on different settings, the quantization method can be viewed as (1) symmetric vs. asymmetric quantization ($Z = 0$ or not), (2) fine-grained vs. coarse-grained quantization (how to partition the input x and get its associated scaling factor, e.g., matrix wise or row wise). See [13] for more details.

Throughout this work, we focus on post-training quantization (PTQ), i.e., no or minimal training effort is applied after quantization, for which large accuracy degradation usually exhibits for coarse-grained quantization (per matrix/tensor) due to their large quantization error. As such, we focus on fine-grained quantization. Particularly, we use the per-row quantization (one row of the weight matrix or one token for the activation) from [36] as our coarsest-grained quantization method, and we use block-k quantization (for every k elements, they have their own scaling factor and/or zero point) as our finer-grained quantization scheme.

## B  Detailed Setting Used in Section 4

Same as [12], for all methods, we use C4 dataset to randomly select 128 sentences for training and each of them has 2048 tokens.

For GPTQ, we check its main hyperparameter, i.e., the dampening factor, and find out the method is not sensitive to it. As such, we use the hyparameter suggested by the author for all of our experiments. For ZQ-Global and ZQ-Local, as mentioned the in main text, the hyperparameters suggested by the original work [36] is suboptimal. We find that a linear decay learning rate schedule is very helpful in our initial test. As such, we add this as our default setting. Meanwhile, we extensively test a wide range (1e-3 to 5e-8) of learning rate for different models until we find the best learning rate (i.e., larger or smaller learning rate leads to worse accuracy performance).We employed the Adam optimizer and set the default batch size to 1 for our experiments.

We conducted tests to assess whether changes in random seeds would introduce substantial variations in the outcomes. As per the findings detailed in Table Table B.1, the modifications in random seeds resulted in only minimal effects on the final quality of the models. This effect was particularly negligible in the context of larger models, such as OPT-30b, where the standard deviation was only 0.01. Therefore, in consideration of these results, we elected to standardize the random seed for the subsequent experiments presented in this paper, setting it uniformly at 123 or 0. The code will be made publicly available to facilitate reproducibility of our results.

For all three methods, we run them on a single GPU (either V100-32GB or A100-80GB). For the largest model tested in the paper, i.e., BLOOM-176B, the cost of all methods is lower than one GPU-day on A100-80G.

Table B.1: The table on the left illustrates the outcomes of each task, evaluated using three different random seeds. On the right, we present a table detailing the mean and standard deviation of the Task-mean values (which can be found in the final column of the left table) over the three random seeds, accompanied by additional quantization results. The quantization methodologies employed in this context are based on the GPTQ algorithm.

| Precision | Random Seed | WikiText | PTB | C4 | Task-mean |
|---|---|---|---|---|---|
| OPT-13b W4A16 | 123 | 10.31 | 12.62 | 11.35 | 11.43 |
| | 234 | 10.25 | 12.57 | 11.35 | 11.39 |
| | 456 | 10.37 | 12.61 | 11.36 | 11.44 |
| OPT-30b W4A16 | 123 | 9.56 | 11.95 | 10.79 | 10.77 |
| | 234 | 9.6 | 11.95 | 10.79 | 10.78 |
| | 456 | 9.52 | 11.97 | 10.79 | 10.76 |

| Precision | Items | OPT-1.3b | OPT-13b | OPT-30b |
|---|---|---|---|---|
| W4A16 | mean over three random seeds | 16.39 | 11.42 | 10.77 |
| | standard deviation | 0.019 | 0.027 | 0.010 |
| W4A8 | mean over three random seeds | 16.76 | 17.16 | 21.64 |
| | standard deviation | 0.048 | 0.048 | 1.277 |

Table C.1: Best optimization method of OPT family in Section 4.

| Precision | 125m | 350m | 1.3b | 2.7b | 6.7b | 13b | 30b | 66b |
|---|---|---|---|---|---|---|---|---|
| Weight Only (INT4) | ZQ-Global | ZQ-Global | GPTQ | GPTQ | GPTQ | GPTQ | GPTQ | GPTQ |
| Weight & Activation (W4A8) | ZQ-Global | ZQ-Global | ZQ-Global | GPTQ | ZQ-Global | ZQ-Global | ZQ-Global | ZQ-Local |

Table C.2: Best optimization method of BLOOM family in Section 4.

| Precision | 560m | 1.1b | 1.7b | 3b | 7.1b | 176b |
|---|---|---|---|---|---|---|
| Weight Only (INT4) | GPTQ | ZQ-Global | ZQ-Global | ZQ-Global/GPTQ | GPTQ | GPTQ |
| Weight & Activation (W4A8) | ZQ-Global | ZQ-Global | ZQ-Global | ZQ-Global | ZQ-Global | ZQ-Local |

## C  Best PTQ Methods with Per-row Quantization

Table C.1 and C.2 summarize the best PTQ methods with per-row optimization.

## D  Quantization-aware training with LoRC

In order to better understand our proposed algorithm, LoRC, particularly in relation to the dimensions of low-rank matrices, we applied quantize-aware training alongside knowledge distillation. This approach builds upon the methodology of row-wise weight quantization and token-wise quantization. For the optimization process, we employed the Adam optimizer, setting the learning rate at 1e-4 and a dropout rate of 0.05. These settings were identified as the most effective in our context (additional details can be found in [33]). We performed fine-tuning on the WikiText dataset using pre-trained GPT2 models with 125M and 350M parameters, which were obtained from Hugging Face as our initial models. [7]

The results are illustrated in Figure Figure D.1. As observed, the quantized models tend to overfit swiftly. However, implementing higher dropout values, such as 0.1, does not result in a significantly improved performance with regards to the best perplexity over the entire training duration. Now when examining the best perplexity associated with each dimension of LoRC (also indicated in the figure's legend), it becomes evident that the larger the dimension, the better the W4A8 models perform. This suggests that augmenting the dimension of LoRC can enhance the model quality for QAT, a finding that deviates from the trends observed in PTQ.

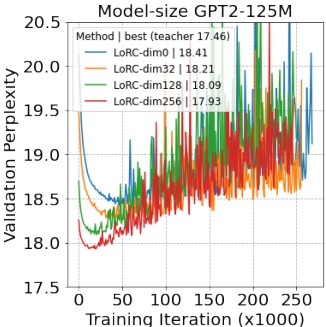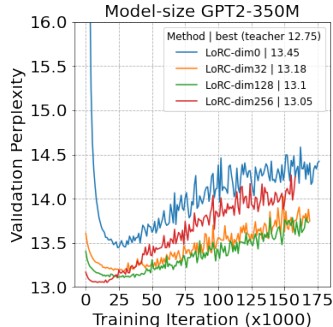

Figure D.1: The graph on the left represents the results for a smaller model size (GPT2-125M), while the one on the right corresponds to the GPT2-350M model. The dimension (refer to the legend) in the LoRC algorithm, which is represented by different color curves, plays a pivotal role in approximating the original quality of the fp16 model.

## E  Tables and Figures

We put the full results of our evaluations in this section.

---

[7] https://huggingface.co/gpt2

Table E.1: OPT ppl on wikitext/ptb/c4 (full results of Table 2).

| Precision | 125m | 350m | 1.3b | 2.7b | 6.7b | 13b | 30b | 66b |
|---|---|---|---|---|---|---|---|---|
| W16-A16 | 27.65/32.55/24.61 | 22.00/26.08/20.71 | 14.62/16.97/14.72 | 12.47/15.11/13.17 | 10.86/13.09/11.74 | 10.13/12.34/11.20 | 9.56/11.84/10.69 | 9.34/11.36/10.28 |
| W8A8$^{sym}$-A16 | 27.64/32.53/24.65 | 22.06/26.10/20.72 | 14.63/16.98/14.73 | 12.48/15.13/13.17 | 10.85/13.11/11.75 | 10.12/12.34/11.20 | 9.55/11.85/10.70 | 9.34/11.36/10.29 |
| W8$^{asym}$-A16 | 27.71/32.58/24.64 | 22.04/26.12/20.73 | 14.67/16.99/14.73 | 12.50/15.14/13.17 | 10.86/13.11/11.75 | 10.11/12.34/11.20 | 9.55/11.84/10.69 | 9.35/11.36/10.29 |
| W4$^{sym}$-A16 | 45.89/53.68/36.68 | 25.95/31.11/23.94 | 19.85/23.61/18.90 | 22.86/30.01/22.29 | 12.41/17.05/13.62 | 11.06/14.90/12.23 | 11.08/13.26/11.06 | 57.73/134.91/98.51 |
| W4$^{asym}$-A16 | 36.71/44.76/30.92 | 25.51/30.90/23.86 | 19.38/21.95/17.93 | 17.92/22.48/18.32 | 11.91/15.39/13.01 | 10.67/13.53/12.07 | 10.10/13.13/11.33 | 20.24/48.45/25.86 |
| W16-A8$^{sym}$ | 27.96/32.57/24.69 | 22.06/26.42/20.95 | 15.21/18.18/15.81 | 12.98/16.01/13.89 | 20.99/25.94/31.18 | 3341.50/2618.38/3554.59 | 1681.48/2221.62/2241.53 | 2696.91/2647.41/2569.94 |
| W16-A8$^{asym}$ | 27.84/32.60/24.66 | 22.04/26.22/20.81 | 15.14/17.65/15.39 | 12.51/15.38/13.38 | 11.24/14.17/12.45 | 11.83/18.87/15.39 | 14.08/31.54/25.09 | 442.66/524.57/716.83 |

Table E.2: BLOOM ppl on wikitext/ptb/c4 (full results of Table **??**).

| Precision | 560m | 1.1b | 1.7b | 3b | 7.1b | 176b |
|---|---|---|---|---|---|---|
| W16-A16 | 22.43/41.25/24.38 | 17.69/46.98/20.29 | 15.39/27.93/17.97 | 13.48/23.12/16.14 | 11.37/19.40/14.13 | 8.11/13.62/10.97 |
| W8$^{sym}$-A16 | 22.44/41.28/24.39 | 17.70/47.01/20.29 | 15.40/27.91/17.98 | 13.49/23.14/16.14 | 11.37/19.40/14.13 | 8.11/13.63/10.98 |
| W8$^{asym}$-A16 | 22.43/41.24/24.40 | 17.69/47.00/20.29 | 15.40/27.91/17.98 | 13.48/23.14/16.14 | 11.37/19.40/14.13 | 8.10/13.62/10.98 |
| W4$^{sym}$-A16 | 26.49/49.73/27.98 | 20.27/56.64/22.81 | 17.47/32.20/19.88 | 14.96/25.59/17.51 | 12.38/21.36/15.06 | 8.40/14.15/11.30 |
| W4$^{asym}$-A16 | 25.31/46.79/27.10 | 23.90/68.31/25.99 | 16.93/31.02/19.47 | 14.65/25.12/17.26 | 12.06/20.83/14.83 | 8.34/14.03/11.23 |
| W16-A8$^{sym}$ | 22.50/41.58/24.46 | 17.78/47.28/20.38 | 15.57/28.36/18.13 | 13.57/23.38/16.25 | 11.58/19.92/14.35 | 8.75/14.94/12.61 |
| W16-A8$^{asym}$ | 22.45/41.37/24.42 | 17.71/47.05/20.32 | 15.45/28.09/18.02 | 13.52/23.24/16.19 | 11.47/19.71/14.25 | 8.41/14.52/11.93 |

Table E.3: OPT ppl on wikitext/opt/c4 with W4$^{asym}$-A16 (full table of Table 3). See Table E.4 for all learning rate results of ZQ-Local and Table E.5 of ZQ-Global.

| Precision | 125m | 350m | 1.3b | 2.7b | 6.7b | 13b | 30b | 66b |
|---|---|---|---|---|---|---|---|---|
| RTN | 36.71/44.76/30.92 | 25.51/30.90/23.86 | 19.38/21.95/17.93 | 17.92/22.48/18.32 | 11.91/15.39/13.01 | 10.67/13.53/12.07 | 10.10/13.13/11.33 | 20.24/48.45/25.86 |
| GPTQ | 32.52/40.25/27.78 | 23.50/29.14/22.41 | 15.52/18.16/15.56 | 13.02/15.84/13.73 | 11.16/13.59/12.08 | 10.29/12.61/11.35 | 9.61/11.95/10.79 | 9.54/11.67/10.52 |
| ZQ-Local* | 33.05/39.34/28.11 | 24.40/29.22/22.82 | 15.81/18.66/15.76 | 13.22/16.19/13.96 | 11.32/13.79/12.26 | 10.42/12.90/11.60 | 9.97/12.32/11.03 | 9.91/11.87/10.59 |
| ZQ-Global* | 31.44/36.66/27.21 | 23.32/28.05/21.98 | 15.46/18.31/15.67 | 13.03/16.04/13.83 | 11.30/13.69/12.17 | 10.38/12.85/11.62 | 9.90/12.24/10.99 | 9.62/11.81/10.61 |

Table E.4: OPT ppl on wikitext/opt/c4 with W4$^{asym}$-A16 and ZQ-Local.

| LR (W4$^{asym}$-A16) | 125m | 350m | 1.3b | 2.7b | 6.7b | 13b | 30b | 66b |
|---|---|---|---|---|---|---|---|---|
| 0.001 | 33.67/39.45/29.11 | 26.33/31.94/24.49 | 16.27/19.91/16.46 | 14.34/17.76/14.93 | 11.87/15.04/13.06 | 13.68/18.89/14.46 | 171.35/151.55/46.14 | 814.22/601.74/308.53 |
| 0.0005 | 32.76/39.51/28.64 | 25.88/30.95/23.96 | 16.29/19.82/16.27 | 14.16/17.65/14.79 | 11.92/15.23/12.95 | 10.93/13.82/12.03 | 10.23/13.46/11.44 | 10.10/12.70/10.81 |
| 0.0001 | 33.86/40.01/28.29 | 24.64/30.26/23.33 | 16.07/19.25/15.93 | 14.36/17.38/14.41 | 11.85/14.64/12.74 | 10.93/13.48/11.88 | 10.18/12.67/11.13 | 10.12/12.01/10.67 |
| 5e-05 | 33.05/39.34/28.11 | 24.40/29.22/22.82 | 15.79/19.16/15.88 | 13.70/16.80/14.16 | 11.71/14.32/12.41 | 10.75/13.38/11.77 | 9.95/12.54/11.09 | 10.02/11.89/10.64 |
| 1e-05 | 33.78/40.41/28.84 | 24.40/29.22/22.82 | 15.81/18.66/15.76 | 13.55/16.46/13.96 | 11.32/13.79/12.26 | 10.54/13.05/11.61 | 9.98/12.22/10.99 | 9.91/11.87/10.59 |
| 5e-06 | 34.47/41.04/29.02 | 24.50/29.27/23.00 | 16.01/18.73/15.91 | 13.22/16.19/13.96 | 11.33/13.86/12.29 | 10.42/12.90/11.60 | 9.86/12.33/10.97 | 9.97/11.86/10.60 |
| 1e-06 | 35.88/43.69/30.35 | 24.54/29.87/23.17 | 16.77/19.45/16.47 | 13.60/17.02/14.46 | 11.41/14.10/12.41 | 10.53/13.01/11.70 | 9.97/12.33/11.04 | 10.01/11.93/10.66 |

Table E.5: OPT ppl on wikitext/opt/c4 with W4$^{asym}$-A16 and ZQ-Global. NaN here means the PPL is larger than 1e6.

| LR (W4$^{asym}$-A16) | 125m | 350m | 1.3b | 2.7b | 6.7b | 13b | 30b | 66b |
|---|---|---|---|---|---|---|---|---|
| 0.001 | 4057.13/2718.91/1247.78 | 5071.35/5229.93/687.35 | 12105.25/10154.73/7893.43 | 18965.76/17112.60/16316.31 | 60014.66/56041.86/78085.84 | 232421.09/98305.32/119762.73 | 93917.09/70170.34/51124.06 | NaN |
| 0.0005 | 31.94/38.61/27.17 | 27.11/33.91/24.07 | 10900.84/8322.65/8425.10 | 14412.30/8676.76/10154.55 | 18527.46/13530.12/13029.95 | 109006.53/62584.41/125349.50 | 303235.75/230599.62/430480.03 | 36439.32/30554.19/37356.93 |
| 0.0001 | 31.44/36.66/27.21 | 24.08/29.08/22.27 | 15.91/20.08/16.35 | 118.38/53.47/54.08 | 7604.92/5339.10/5161.49 | 12638.86/7639.95/8243.63 | 16276.68/9890.26/6176.27 | 8367.31/4728.13/5533.59 |
| 5e-05 | 31.97/36.93/27.12 | 23.55/29.02/22.02 | 15.82/18.65/15.65 | 13.40/16.44/13.97 | 26.54/25.67/17.60 | 909.99/316.82/370.84 | 6238.21/5291.04/3743.01 | 9296.98/6687.44/5363.29 |
| 1e-05 | 32.31/37.93/27.38 | 23.32/28.05/21.98 | 15.60/18.42/15.64 | 13.09/16.05/13.78 | 11.41/13.82/12.20 | 10.80/13.16/11.66 | 10.06/12.44/11.07 | 9.73/12.09/10.98 |
| 5e-06 | 32.69/38.91/27.76 | 23.26/28.33/22.05 | 15.46/18.31/15.67 | 13.03/16.04/13.83 | 11.30/13.69/12.17 | 10.50/12.89/11.58 | 9.95/12.28/11.01 | 9.62/11.81/10.61 |
| 1e-06 | 34.63/41.75/29.43 | 23.82/28.96/22.48 | 16.12/19.46/16.27 | 13.03/16.27/14.04 | 11.29/13.88/12.27 | 10.38/12.85/11.62 | 9.90/12.24/10.99 | 9.58/12.17/10.78 |
| 5e-07 | NaN | NaN | NaN | NaN | NaN | 10.51/12.96/11.70 | 9.89/12.41/11.04 | 9.90/12.45/11.00 |
| 1e-07 | NaN | NaN | NaN | NaN | NaN | 10.63/13.29/11.89 | 10.02/12.82/11.18 | 11.03/13.91/11.73 |
| 5e-08 | NaN | NaN | NaN | NaN | NaN | 10.66/13.42/11.97 | 10.05/13.00/11.24 | 12.41/17.45/13.02 |

Table E.6: BLOOM ppl on wikitext/opt/c4 with W4$^{asym}$-A16 (full table of Table 3). See Table E.4 for all learning rate results of ZQ-Local and Table E.5 of ZQ-Global.

| Precision | 560m | 1.1b | 1.7b | 3b | 7.1b | 176b |
|---|---|---|---|---|---|---|
| RTN | 25.31/46.79/27.10 | 23.90/68.31/25.99 | 16.93/31.02/19.47 | 14.65/25.12/17.26 | 12.06/20.83/14.83 | 8.34/14.03/11.23 |
| GPTQ | 23.90/43.76/25.59 | 24.34/68.10/26.58 | 16.36/29.58/18.79 | 14.10/24.23/16.66 | 11.80/20.23/14.47 | 8.22/13.78/11.07 |
| ZQ-Local* | 24.23/44.94/26.05 | 19.22/52.36/21.59 | 16.37/29.89/18.86 | 14.23/24.41/16.86 | 11.80/20.28/14.56 | 8.27/13.91/11.16 |
| ZQ-Global* | 23.84/44.17/25.60 | 19.50/51.33/21.72 | 16.19/29.28/18.66 | 14.14/24.16/16.69 | 11.77/20.27/14.52 | 8.24/13.82/11.10 |

Table E.7: BLOOM ppl on wikitext/opt/c4 with W4$^{asym}$-A16 and ZQ-Local.

| LR (W4$^{asym}$-A16) | 560m | 1.1b | 1.7b | 3b | 7.1b | 176b |
|---|---|---|---|---|---|---|
| 0.001 | 25.37/47.36/27.03 | 19.89/53.86/22.11 | 16.70/31.19/19.30 | 14.45/25.28/17.16 | 12.22/21.34/15.04 | 8.82/15.77/11.98 |
| 0.0005 | 25.17/46.83/26.87 | 19.57/53.66/21.92 | 16.58/30.27/19.15 | 14.43/25.47/17.07 | 11.94/20.54/14.67 | 8.35/14.01/11.20 |
| 0.0001 | 24.59/46.11/26.32 | 19.22/52.36/21.59 | 16.41/30.29/18.90 | 14.35/24.81/16.87 | 11.80/20.28/14.58 | 8.28/13.92/11.14 |
| 5e-05 | 24.44/46.04/26.16 | 23.28/65.68/25.42 | 16.39/30.01/18.86 | 14.34/24.43/16.83 | 11.80/20.28/14.56 | 8.27/13.93/11.15 |
| 1e-05 | 24.23/44.94/26.05 | 23.45/66.29/25.52 | 16.37/29.89/18.86 | 14.23/24.41/16.86 | 11.84/20.39/14.58 | 8.27/13.91/11.16 |
| 5e-06 | 24.21/45.21/26.10 | 23.26/65.72/25.42 | 16.42/30.09/18.94 | 14.25/24.55/16.87 | 11.87/20.50/14.61 | 8.29/13.98/11.16 |
| 1e-06 | 24.71/45.86/26.50 | 23.45/66.28/25.56 | 16.64/30.52/19.15 | 14.46/24.76/17.04 | 11.94/20.55/14.70 | 8.29/13.97/11.18 |

Table E.8: BLOOM ppl on wikitext/opt/c4 with W4$^{asym}$-A16 and ZQ-Global.

| LR (W4$^{asym}$-A16) | 560m | 1.1b | 1.7b | 3b | 7.1b | 176b |
|---|---|---|---|---|---|---|
| 0.001 | 6853935.00/30441738.00/3222857.25 | 528072.88/828428.62/356031.97 | 597410.50/973155.88/1280478.12 | 878460.69/2175974.25/441401.94 | nan/nan/nan | NaN |
| 0.0005 | 29671.52/179030.88/4653.35 | 28112.96/87515.64/1826.82 | 141110.14/204295.86/40146.11 | 265457.25/741326.38/99882.45 | 944784.19/774538.25/395960.03 | NaN |
| 0.0001 | 23.92/45.68/25.72 | 19.34/52.78/21.63 | 16.35/29.22/18.56 | 14.27/24.46/16.80 | 12.17/22.16/14.80 | NaN |
| 5e-05 | 23.84/44.17/25.60 | 19.50/51.33/21.72 | 16.19/29.28/18.66 | 14.14/24.16/16.69 | 11.81/20.41/14.50 | NaN |
| 1e-05 | 23.85/44.20/25.65 | 22.64/56.79/23.41 | 16.23/29.73/18.73 | 14.14/24.31/16.74 | 11.77/20.27/14.52 | 8.24/13.82/11.10 |
| 5e-06 | 24.02/44.62/25.79 | 23.46/63.27/24.88 | 16.28/29.83/18.81 | 14.19/24.38/16.80 | 11.77/20.33/14.54 | 8.24/13.82/11.10 |
| 1e-06 | 24.46/45.41/26.20 | 24.62/70.16/26.64 | 16.48/30.15/19.02 | 14.35/24.56/16.95 | 11.89/20.54/14.67 | 8.23/13.82/11.12 |
| 5e-07 | NaN | NaN | NaN | NaN | NaN | 8.26/13.86/11.13 |

Table E.9: OPT ppl on wikitext/opt/c4 with W4$^{\text{asym}}$-A8$^{\text{sym}}$/A8$^{\text{asym}}$. See Table E.10 for all learning rate results of ZQ-Local and Table E.11 of ZQ-Global.

| Precision | 125m | 350m | 1.3b | 2.7b | 6.7b | 13b | 30b | 66b |
|---|---|---|---|---|---|---|---|---|
| W4$^{\text{asym}}$-A8$^{\text{sym}}$ Block | | | | | | | | |
| RTN | 36.69/44.34/30.60 | 26.59/32.13/24.81 | 25.31/26.89/22.01 | 30.84/35.73/29.01 | 164.51/110.85/162.94 | 4460.61/3145.51/4255.84 | 3216.45/2929.40/3570.19 | 3038.22/2930.92/3001.82 |
| GPTQ | 32.20/38.49/27.47 | 24.35/29.82/23.24 | 16.28/19.64/16.73 | 13.86/17.51/15.00 | 46.22/53.98/55.13 | 3611.71/2796.71/3820.57 | 1738.44/1810.08/2119.82 | 5992.87/4115.01/4360.16 |
| ZQ-Local* | 32.88/38.23/28.20 | 25.18/30.06/23.62 | 16.78/20.25/17.09 | 14.82/18.77/15.61 | 16.08/21.15/18.77 | 2680.33/1876.48/3052.51 | 1884.90/1603.23/1348.08 | 575.20/499.42/437.94 |
| ZQ-Global* | 32.04/37.48/27.23 | 24.01/28.81/22.57 | 16.12/19.15/16.23 | 13.98/17.70/14.87 | 38.27/39.77/52.26 | 117.83/141.63/96.83 | 253.71/700.40/337.15 | 1715.98/1546.50/1799.35 |
| W4$^{\text{asym}}$-A8$^{\text{asym}}$ Block | | | | | | | | |
| RTN | 36.61/44.48/30.64 | 25.79/31.28/24.13 | 21.23/23.54/19.19 | 23.82/29.77/22.60 | 13.18/17.04/14.19 | 19.87/32.93/26.28 | 36.07/136.88/85.84 | 627.15/880.79/937.08 |
| GPTQ | 32.22/38.83/27.43 | 23.90/29.29/22.63 | 15.75/18.74/15.93 | 13.23/16.31/14.03 | 12.50/15.86/13.29 | 12.79/21.99/17.05 | 12.96/25.03/24.14 | 495.70/681.68/768.69 |
| ZQ-Local* | 33.60/38.57/28.02 | 24.57/29.27/22.98 | 15.98/19.13/16.20 | 13.44/16.81/14.26 | 11.76/14.97/13.00 | 11.69/16.98/14.01 | 12.38/24.25/18.96 | 12.19/23.31/13.47 |
| ZQ-Global* | 31.61/37.00/27.10 | 23.66/28.56/22.21 | 15.77/18.61/15.83 | 13.09/16.56/14.00 | 12.03/14.60/12.86 | 11.80/15.01/12.41 | 12.94/17.61/13.41 | 31.51/58.00/23.95 |

Table E.10: OPT ppl on wikitext/opt/c4 with W4$^{\text{asym}}$-A8$^{\text{sym}}$/A8$^{\text{asym}}$ and ZQ-Local.

| Precision | 125m | 350m | 1.3b | 2.7b | 6.7b | 13b | 30b | 66b |
|---|---|---|---|---|---|---|---|---|
| W4$^{\text{asym}}$-A8$^{\text{sym}}$ Block | | | | | | | | |
| 0.001 | 34.91/40.43/29.37 | 26.82/32.68/25.24 | 17.68/21.72/18.11 | 19.40/27.59/20.05 | 36.70/59.32/45.17 | 7240.89/5506.67/4889.34 | 8229.32/5068.14/5005.13 | Diverge |
| 0.0005 | 34.16/39.00/28.58 | 26.75/32.05/24.60 | 17.19/21.42/17.55 | 19.43/25.54/19.41 | 29.33/48.38/43.28 | 56836.57/56810.64/31073.67 | 5448.96/3826.63/3196.49 | 575.20/499.42/437.94 |
| 0.0001 | 32.88/38.23/28.20 | 25.31/31.60/23.98 | 16.93/20.77/17.36 | 17.05/21.50/17.42 | 25.24/31.66/26.82 | 6125.07/3817.01/4121.70 | 1884.90/1603.23/1348.08 | 5427.12/3449.58/3289.01 |
| 5e-05 | 32.86/39.17/27.91 | 25.91/31.24/24.07 | 16.99/20.02/17.23 | 15.07/19.00/15.54 | 16.08/21.15/18.77 | 6037.51/3617.64/3819.63 | 3266.46/2533.64/2463.21 | 11631.78/10489.81/7880.43 |
| 1e-05 | 34.00/39.76/28.62 | 25.40/30.60/23.75 | 16.87/20.26/17.11 | 14.82/18.77/15.61 | 26.60/32.09/28.76 | 5346.85/3788.29/4903.31 | 3364.70/2372.71/3370.97 | 5793.44/3544.90/3925.34 |
| 5e-06 | 34.37/41.46/28.71 | 25.18/30.06/23.62 | 16.78/20.25/17.09 | 14.87/19.42/15.86 | 34.53/39.98/38.22 | 2680.33/1876.48/3052.51 | 3566.45/2532.54/3678.75 | 4916.96/3783.69/3716.49 |
| 1e-06 | 36.05/43.46/30.00 | 25.73/30.69/24.05 | 19.58/22.57/19.04 | 18.66/24.19/19.98 | 77.99/62.27/83.19 | 3893.00/2672.11/3849.59 | 3233.72/2944.44/3732.18 | 4238.57/3621.09/3541.33 |
| W4$^{\text{asym}}$-A8$^{\text{asym}}$ Block | | | | | | | | |
| 0.001 | 33.57/40.84/29.00 | 27.29/32.48/24.68 | 17.41/20.70/17.07 | 15.98/20.45/16.23 | 12.63/17.21/14.25 | 9889.96/7605.54/6328.91 | 2009.66/1637.69/2011.15 | 5070.07/3124.56/2683.19 |
| 0.0005 | 34.58/40.45/28.69 | 25.81/31.56/24.09 | 16.89/20.66/16.93 | 15.00/19.47/15.61 | 12.55/17.00/14.29 | 13.18/19.65/15.18 | 36.51/75.89/60.58 | 3249.10/63.17/119.55 |
| 0.0001 | 33.91/38.39/28.12 | 25.37/31.24/23.66 | 16.78/20.09/16.72 | 14.26/18.49/14.90 | 12.13/15.97/13.48 | 13.48/20.42/16.68 | 110.20/117.28/257.96 | 12.19/23.31/13.47 |
| 5e-05 | 33.60/38.57/28.02 | 24.67/29.60/23.34 | 16.31/19.56/16.42 | 13.90/19.16/15.05 | 12.30/15.95/13.56 | 12.05/18.00/15.77 | 37.68/59.83/124.75 | 29.72/95.99/69.60 |
| 1e-05 | 33.80/40.21/28.56 | 24.57/29.27/22.98 | 15.98/19.13/16.20 | 13.44/16.81/14.26 | 11.76/14.97/13.00 | 11.69/16.98/14.01 | 14.39/31.47/24.45 | 217.93/313.13/298.24 |
| 5e-06 | 34.62/41.07/28.93 | 24.68/29.46/23.12 | 16.26/19.23/16.27 | 13.44/17.00/14.36 | 11.96/14.86/13.10 | 12.31/18.55/15.16 | 12.38/24.25/18.96 | 85.96/185.07/180.88 |
| 1e-06 | 35.94/43.35/30.00 | 24.92/30.18/23.45 | 17.98/20.89/17.45 | 14.79/18.90/15.52 | 12.10/15.47/13.35 | 15.48/22.00/17.84 | 14.86/31.16/26.21 | 411.89/620.52/652.55 |

Table E.11: OPT ppl on wikitext/opt/c4 with W4$^{\text{asym}}$-A8$^{\text{sym}}$/A8$^{\text{asym}}$ and ZQ-Global.

| Precision | 125m | 350m | 1.3b | 2.7b | 6.7b | 13b | 30b | 66b |
|---|---|---|---|---|---|---|---|---|
| W4$^{\text{asym}}$-A8$^{\text{sym}}$ Block | | | | | | | | |
| 0.001 | 34.90/44.82/28.27 | 8988.08/5862.33/384.69 | nan/nan/nan | 18290.16/9784.37/12099.01 | 16014.50/8655.69/12304.55 | 248961.98/84832.78/104880.55 | 56675.05/23709.03/33007.17 | 29782.43/20410.10/23559.66 |
| 0.0005 | 31.78/38.56/27.20 | 39.24/54.15/29.76 | 10610.96/9438.99/6752.84 | 12499.29/8411.26/10677.01 | nan/nan/nan | 74731.13/44494.68/29286.49 | 51871.73/28548.95/23056.78 | 18717.63/17144.97/12903.33 |
| 0.0001 | 32.04/37.48/27.23 | 24.14/29.21/22.47 | 17.04/23.64/17.13 | 175.67/165.81/162.24 | 12305.50/11472.90/10223.89 | 16303.04/10731.12/10669.52 | 22548.81/12474.28/7405.46 | 7926.43/4377.36/4805.98 |
| 5e-05 | 32.16/37.54/27.27 | 24.15/28.37/22.46 | 16.02/19.61/16.59 | 13.88/20.27/14.79 | 5241.10/3284.47/2187.15 | 13297.25/7781.85/7467.30 | 9542.44/4543.45/5373.00 | NaN |
| 1e-05 | 32.57/38.43/27.53 | 24.01/28.81/22.57 | 16.12/19.15/16.23 | 13.98/17.70/14.87 | 99.27/118.19/88.74 | 529.82/361.44/256.46 | 1936.12/1368.68/947.45 | 10077.70/9208.34/11462.28 |
| 5e-06 | 32.83/38.37/27.71 | 24.13/29.30/22.68 | 16.45/19.64/16.57 | 14.42/18.01/15.27 | 70.26/62.28/54.47 | 373.82/494.33/170.40 | 820.90/847.19/543.59 | 1867.57/1878.76/4117.49 |
| 1e-06 | 34.79/41.79/29.30 | 24.68/30.01/23.23 | 17.90/21.94/18.01 | 14.83/18.63/15.70 | 38.27/39.77/52.26 | 117.83/141.63/96.83 | 261.19/844.40/272.04 | 1500.51/1275.54/1649.50 |
| 5e-07 | NaN | NaN | NaN | NaN | NaN | NaN | 253.71/700.40/337.15 | 1715.98/1546.50/1799.35 |
| 1e-07 | NaN | NaN | NaN | NaN | NaN | NaN | 913.95/1117.58/1065.87 | 2012.91/1917.48/1817.92 |
| W4$^{\text{asym}}$-A8$^{\text{asym}}$ Block | | | | | | | | |
| 0.001 | 37.89/47.68/30.43 | 9023.01/4309.50/1186.96 | 12638.86/nan/9164.64 | 11285.86/6477.19/nan | 12222.01/6933.34/8989.30 | 132962.69/73768.05/59268.76 | 328993.91/187752.97/163157.59 | 48298.52/30548.89/42797.96 |
| 0.0005 | 32.65/39.82/27.20 | 28.46/36.94/24.68 | nan/nan/nan | nan/nan/nan | 23287.96/15508.32/16243.28 | 22052.30/10852.90/11588.02 | 63084.59/39919.41/42499.90 | NaN |
| 0.0001 | 31.61/37.00/27.10 | 24.64/29.13/22.28 | 16.31/19.71/16.44 | 43.76/29.11/33.35 | 22024.01/13962.04/14130.94 | 10171.49/7200.78/7954.12 | 18603.08/11639.42/10798.26 | nan/nan/nan |
| 5e-05 | 32.21/37.46/27.18 | 23.66/28.56/22.21 | 16.02/19.02/15.92 | 13.48/17.57/14.24 | 839.48/213.76/286.05 | 1035.13/nan/1472.00 | 8085.92/3545.21/4893.07 | nan/nan/nan |
| 1e-05 | 32.35/38.21/27.38 | 23.59/28.66/22.24 | 15.77/18.61/15.83 | 13.09/16.56/14.00 | 12.09/14.69/12.90 | 11.80/15.01/12.41 | 13.76/22.87/15.72 | 974.58/1557.95/1039.65 |
| 5e-06 | 32.59/38.49/27.68 | 23.62/28.63/22.33 | 15.78/18.80/15.95 | 13.23/16.65/14.12 | 12.03/14.60/12.86 | 12.72/16.31/13.20 | 12.94/17.61/13.41 | 83.35/137.83/128.11 |
| 1e-06 | 34.68/41.56/29.26 | 24.08/29.21/22.68 | 16.66/20.03/16.69 | 13.30/16.74/14.33 | 12.43/15.52/13.36 | 12.28/16.13/13.19 | 16.00/19.60/14.88 | 31.51/58.00/23.95 |
| 5e-07 | NaN | NaN | NaN | NaN | NaN | NaN | NaN | 31.09/73.23/24.44 |
| 1e-07 | NaN | NaN | NaN | NaN | NaN | NaN | NaN | 241.81/544.81/505.58 |

Table E.12: BLOOM ppl on wikitext/opt/c4 with W4$^{\text{asym}}$-A8$^{\text{sym}}$/A8$^{\text{asym}}$. See Table E.13 for all learning rate results of ZQ-Local and Table E.14 of ZQ-Global.

| Precision | 560m | 1.1b | 1.7b | 3b | 7.1b | 176b |
|---|---|---|---|---|---|---|
| W4$^{\text{asym}}$-A8$^{\text{sym}}$ Block | | | | | | |
| RTN | 25.56/47.53/27.31 | 24.80/70.99/26.71 | 17.36/31.95/19.89 | 14.82/25.63/17.47 | 12.33/21.62/15.13 | 9.12/15.58/14.04 |
| GPTQ | 24.13/44.79/25.86 | 25.69/68.65/27.08 | 16.63/30.54/19.12 | 14.18/24.42/16.82 | 12.04/21.07/14.75 | 8.92/15.16/13.56 |
| ZQ-Local* | 24.45/45.73/26.22 | 19.50/52.67/21.73 | 16.71/30.23/19.09 | 14.37/24.72/16.99 | 12.00/20.79/14.78 | 8.52/14.29/11.41 |
| ZQ-Global* | 23.93/44.31/25.68 | 19.71/51.98/21.85 | 16.34/29.36/18.82 | 14.13/24.34/16.76 | 11.84/20.58/14.59 | 8.76/14.60/11.68 |
| W4$^{\text{asym}}$-A8$^{\text{asym}}$ Block | | | | | | |
| RTN | 25.37/46.99/27.16 | 24.08/68.95/26.17 | 17.12/31.46/19.67 | 14.74/25.38/17.37 | 12.22/21.36/15.00 | 8.73/15.10/12.83 |
| GPTQ | 24.09/44.29/25.66 | 24.50/67.37/26.62 | 16.39/29.83/18.91 | 14.13/24.47/16.73 | 11.91/20.72/14.62 | 8.55/14.74/12.31 |
| ZQ-Local* | 24.29/45.19/26.10 | 19.13/52.89/21.63 | 16.54/30.11/18.92 | 14.32/24.73/16.94 | 11.94/20.63/14.68 | 8.33/14.01/11.22 |
| ZQ-Global* | 23.86/44.16/25.62 | 19.54/51.72/21.79 | 16.23/29.40/18.68 | 14.15/24.29/16.72 | 11.80/20.37/14.56 | 8.62/14.40/11.49 |

Table E.13: BLOOM ppl on wikitext/opt/c4 with W4$^{\text{asym}}$-A8$^{\text{sym}}$/A8$^{\text{asym}}$ and ZQ-Local.

| Precision | 560m | 1.1b | 1.7b | 3b | 7.1b | 176b |
|---|---|---|---|---|---|---|
| W4$^{\text{asym}}$-A8$^{\text{sym}}$ Block | | | | | | |
| 0.001 | 25.51/47.89/27.15 | 19.73/54.63/22.18 | 16.96/31.47/19.44 | 14.59/25.69/17.32 | 12.51/21.85/15.34 | 8.62/14.42/11.50 |
| 0.0005 | 25.18/47.35/26.95 | 19.62/53.64/22.03 | 16.98/31.75/19.47 | 14.52/25.22/17.18 | 12.03/21.01/14.82 | 8.59/14.38/11.45 |
| 0.0001 | 24.79/46.37/26.44 | 19.50/52.67/21.73 | 16.68/30.51/19.18 | 14.44/25.12/17.05 | 12.00/20.79/14.78 | 8.52/14.29/11.41 |
| 5e-05 | 24.56/46.29/26.34 | 23.93/69.17/26.19 | 16.71/30.23/19.09 | 14.37/24.72/16.99 | 12.05/20.92/14.82 | 8.55/14.34/11.44 |
| 1e-05 | 24.45/45.73/26.22 | 23.65/66.73/25.80 | 16.66/30.69/19.16 | 14.40/24.94/17.02 | 12.12/21.14/14.86 | 8.65/14.97/12.01 |
| 5e-06 | 24.48/45.66/26.33 | 23.87/67.26/25.84 | 16.78/30.72/19.23 | 14.44/24.91/17.07 | 12.15/21.23/14.88 | 8.70/15.04/12.37 |
| 1e-06 | 24.91/46.35/26.72 | 24.09/68.13/26.05 | 17.03/31.28/19.52 | 14.60/25.18/17.24 | 12.22/21.31/14.99 | 8.91/15.25/13.35 |
| W4$^{\text{asym}}$-A8$^{\text{asym}}$ Block | | | | | | |
| 0.001 | 25.26/46.43/26.98 | 19.69/54.26/22.14 | 16.88/32.16/19.40 | 15.15/26.58/17.76 | 12.40/22.29/15.28 | 8.40/14.06/11.26 |
| 0.0005 | 24.89/47.99/26.82 | 19.54/53.57/21.98 | 16.73/31.02/19.29 | 14.50/25.52/17.11 | 11.94/20.70/14.76 | 8.33/14.01/11.22 |
| 0.0001 | 24.60/45.75/26.44 | 19.13/52.89/21.63 | 16.54/30.36/19.10 | 14.37/24.91/16.93 | 11.94/20.63/14.68 | 8.35/14.04/11.24 |
| 5e-05 | 24.41/45.08/26.23 | 23.59/67.14/25.79 | 16.54/30.11/18.92 | 14.29/24.83/16.92 | 11.95/20.71/14.71 | 8.36/14.10/11.25 |
| 1e-05 | 24.29/45.19/26.10 | 23.35/65.26/25.38 | 16.51/30.20/19.00 | 14.32/24.73/16.94 | 11.97/20.93/14.74 | 8.44/14.30/11.45 |
| 5e-06 | 24.31/45.25/26.15 | 23.41/66.18/25.48 | 16.63/30.37/19.09 | 14.33/24.74/16.96 | 12.03/20.95/14.78 | 8.52/14.66/11.86 |
| 1e-06 | 24.76/45.92/26.62 | 23.52/66.38/25.66 | 16.81/30.71/19.30 | 14.53/24.92/17.14 | 12.10/21.07/14.87 | 8.62/14.92/12.41 |

Table E.14: BLOOM ppl on wikitext/opt/c4 with W4$^{asym}$-A8$^{sym}$/A8$^{asym}$ and ZQ-Global.

| Precision | 560m | 1.1b | 1.7b | 3b | 7.1b | 176b |
|---|---|---|---|---|---|---|
| W4$^{asym}$-A8$^{sym}$ Block | | | | | | |
| 0.001 | 174250016.00/201477664.00/1348168.88 | 423532.56/906908.06/322995.69 | 573201.81/1089364.38/498071.91 | 544376.56/696942.56/540949.06 | nan/nan/nan | NaN |
| 0.0005 | 70978.52/29214230.00/1151.72 | 2880.81/15732.60/309.13 | 505479.44/629035.56/29283.36 | 140595.53/181082.25/33785.79 | 378033.53/789890.00/191543.91 | NaN |
| 0.0001 | 24.04/45.38/25.83 | 19.44/52.38/21.77 | 16.34/29.36/18.82 | 14.32/24.74/16.88 | 12.12/22.00/14.80 | 249.47/26690.76/26.96 |
| 5e-05 | 23.93/44.31/25.68 | 19.71/51.98/21.85 | 16.18/29.71/18.71 | 14.13/24.34/16.76 | 11.84/20.58/14.59 | 9.00/15.57/11.61 |
| 1e-05 | 23.99/44.44/25.77 | 22.75/58.31/23.63 | 16.28/29.96/18.81 | 14.29/24.53/16.87 | 11.87/20.57/14.64 | 8.76/14.60/11.68 |
| 5e-06 | 24.14/44.77/25.90 | 23.90/64.81/25.29 | 16.36/30.03/18.91 | 14.32/24.68/16.95 | 11.91/20.60/14.71 | 9.07/15.12/11.98 |
| 1e-06 | 24.62/45.70/26.33 | 25.55/71.49/27.44 | 16.61/30.47/19.17 | 14.51/24.91/17.11 | 12.06/20.93/14.86 | 11.25/19.93/15.76 |
| W4$^{asym}$-A8$^{asym}$ Block | | | | | | |
| 0.001 | 9059092.00/2932002.50/131873960.00 | 499829.19/393190.53/346682.47 | 1260531.12/2019747.88/460627.16 | 1022130.19/872164.88/679662.62 | nan/nan/nan | NaN |
| 0.0005 | 7633.14/378055.53/1032.16 | 4271.83/85847.50/1555.66 | 87087.04/217513.30/37000.13 | 575008.56/814032.50/230285.80 | 1212241.00/2389840.25/1504266.50 | NaN |
| 0.0001 | 23.96/45.36/25.80 | 19.37/52.25/21.88 | 16.29/29.36/18.81 | 14.32/24.66/16.86 | 12.05/22.30/14.77 | 1400.84/11880.12/392.79 |
| 5e-05 | 23.86/44.16/25.62 | 19.54/51.72/21.79 | 16.23/29.40/18.68 | 14.15/24.29/16.72 | 11.82/20.44/14.54 | 8.73/20.30/11.41 |
| 1e-05 | 23.96/44.24/25.72 | 22.55/58.10/23.49 | 16.27/29.82/18.78 | 14.16/24.35/16.80 | 11.80/20.37/14.56 | 8.62/14.40/11.49 |
| 5e-06 | 24.01/44.68/25.83 | 23.67/64.20/25.08 | 16.30/29.96/18.85 | 14.24/24.49/16.86 | 11.81/20.50/14.60 | 8.69/14.56/11.58 |
| 1e-06 | 24.53/45.60/26.26 | 24.82/71.17/26.84 | 16.55/30.35/19.10 | 14.40/24.76/17.01 | 11.97/20.83/14.77 | 9.14/16.63/17.69 |

Table E.15: OPT full results of Table 4.

| Method | 125m | 350m | 1.3b | 2.7b | 6.7b | 13b | 30b | 66b |
|---|---|---|---|---|---|---|---|---|
| BS=1024 | | | | | | | | |
| RTN | N/A | 25.42/30.62/23.61 | 16.90/19.78/16.59 | N/A | 11.63/14.41/12.65 | 10.47/13.09/11.75 | 9.97/12.40/11.09 | 9.83/12.31/10.77 |
| | N/A | 26.55 | 17.76 | N/A | 12.90 | 11.77 | 11.15 | 10.97 |
| GPTQ | N/A | 23.65/29.09/22.43 | 15.16/18.00/15.34 | N/A | 11.10/13.40/11.99 | 10.28/12.49/11.29 | 9.58/11.91/10.75 | 9.56/11.61/10.44 |
| | N/A | 25.05 | 16.17 | N/A | 12.16 | 11.36 | 10.75 | 10.54 |
| ZQ-Global* | N/A | 23.27/27.97/21.93 | 12.93/15.90/13.64 | N/A | 10.98/13.60/12.04 | 10.33/12.69/11.50 | 9.78/12.16/10.90 | 9.52/11.58/10.46 |
| | N/A | 24.39 | 16.18 | N/A | 12.21 | 11.50 | 10.95 | 10.52 |
| BS=512 | | | | | | | | |
| RTN | N/A | 25.05/29.74/23.21 | 15.71/19.05/16.09 | 13.67/16.93/14.23 | 11.32/14.22/12.50 | 10.45/12.99/11.68 | 10.03/12.27/11.03 | 9.83/12.15/10.67 |
| | N/A | 26.00 | 16.95 | 14.94 | 12.68 | 11.71 | 11.11 | 10.89 |
| GPTQ | N/A | 23.33/28.48/22.13 | 15.15/17.95/15.26 | 12.65/15.61/13.53 | 10.94/13.37/11.94 | 10.18/12.49/11.29 | 9.58/11.87/10.75 | 9.53/11.59/10.43 |
| | N/A | 24.65 | 16.12 | 13.93 | 12.08 | 11.32 | 10.73 | 10.52 |
| ZQ-Global* | N/A | 23.41/27.67/21.92 | 14.91/17.73/15.25 | 12.92/15.59/13.55 | 11.08/13.51/11.99 | 10.29/12.68/11.46 | 9.79/12.16/10.87 | 9.51/11.65/10.44 |
| | N/A | 24.34 | 15.97 | 14.02 | 12.19 | 11.48 | 10.94 | 10.53 |
| BS=256 | | | | | | | | |
| RTN | 31.62/38.19/27.62 | 24.76/29.44/22.96 | 15.54/18.96/15.90 | 13.56/16.62/14.02 | 11.19/14.12/12.40 | 10.39/12.93/11.61 | 9.95/12.24/10.98 | 9.70/12.09/10.62 |
| | 32.48 | 25.72 | 16.80 | 14.73 | 12.57 | 11.64 | 11.06 | 10.80 |
| GPTQ | 30.56/37.20/26.68 | 23.37/28.33/21.97 | 14.95/17.63/15.16 | 12.59/15.60/13.49 | 10.93/13.29/11.92 | 10.15/12.43/11.27 | 9.58/11.91/10.74 | 9.49/11.60/10.40 |
| | 31.48 | 24.56 | 15.91 | 13.89 | 12.05 | 11.28 | 10.74 | 10.50 |
| ZQ-Global* | 30.45/35.35/26.24 | 23.06/27.72/21.74 | 14.93/17.45/15.15 | 12.99/15.47/13.50 | 10.96/13.45/12.00 | 10.25/12.61/11.43 | 9.73/12.14/10.89 | 9.49/11.58/10.42 |
| | 30.68 | 24.17 | 15.84 | 13.99 | 12.14 | 11.43 | 10.92 | 10.50 |
| BS=128 | | | | | | | | |
| RTN | 30.62/36.67/27.10 | 24.12/29.34/22.70 | 15.35/18.52/15.66 | 13.19/16.24/13.88 | 11.11/13.94/12.28 | 10.31/12.82/11.54 | 9.93/12.12/10.93 | 9.56/11.85/10.56 |
| | 31.47 | 25.39 | 16.51 | 14.43 | 12.44 | 11.56 | 11.00 | 10.65 |
| GPTQ | 30.76/36.13/26.52 | 23.29/27.94/21.98 | 14.93/17.51/15.10 | 12.49/15.59/13.46 | 10.87/13.34/11.90 | 10.11/12.47/11.27 | 9.60/11.88/10.73 | 9.44/11.53/10.40 |
| | 31.14 | 24.40 | 15.85 | 13.85 | 12.03 | 11.28 | 10.74 | 10.45 |
| ZQ-Global* | 29.52/34.63/25.98 | 22.78/27.56/21.65 | 15.02/17.50/15.07 | 12.67/15.37/13.45 | 10.92/13.42/11.96 | 10.16/12.61/11.41 | 9.74/12.01/10.82 | 9.43/11.49/10.40 |
| | 30.04 | 23.99 | 15.86 | 13.83 | 12.10 | 11.39 | 10.86 | 10.44 |
| BS=64 | | | | | | | | |
| RTN | 30.74/36.68/26.87 | 24.28/28.95/22.59 | 15.21/18.15/15.47 | 13.20/16.13/13.75 | 11.01/13.71/12.17 | 10.27/12.79/11.49 | 9.82/12.05/10.89 | 9.46/11.70/10.49 |
| | 31.43 | 25.27 | 16.28 | 14.36 | 12.30 | 11.52 | 10.92 | 10.55 |
| GPTQ | 30.25/35.72/26.43 | 23.39/27.55/21.75 | 14.81/17.40/15.06 | 12.54/15.54/13.44 | 10.87/13.29/11.89 | 10.09/12.44/11.27 | 9.55/11.89/10.72 | 9.33/11.49/10.38 |
| | 30.80 | 24.23 | 15.76 | 13.84 | 12.02 | 11.27 | 10.72 | 10.40 |
| ZQ-Global* | 29.69/34.24/25.72 | 22.94/27.49/21.54 | 14.90/17.43/15.01 | 12.80/15.47/13.44 | 10.92/13.33/11.93 | 10.21/12.58/11.38 | 9.69/12.01/10.87 | 9.41/11.49/10.39 |
| | 29.88 | 23.99 | 15.78 | 13.90 | 12.06 | 11.39 | 10.84 | 10.43 |
| BS=32 | | | | | | | | |
| RTN | 30.48/36.32/26.64 | 23.88/28.66/22.36 | 14.99/17.87/15.32 | 12.89/16.00/13.67 | 10.89/13.70/12.13 | 10.32/12.73/11.45 | 9.76/12.00/10.85 | 9.56/11.55/10.44 |
| | 31.14 | 24.97 | 16.06 | 14.18 | 12.24 | 11.50 | 10.87 | 10.52 |
| GPTQ | 29.13/34.89/25.90 | 23.09/27.59/21.65 | 14.80/17.41/15.04 | 12.45/15.55/13.42 | 10.89/13.32/11.89 | 10.08/12.48/11.27 | 9.51/11.92/10.73 | Diverge |
| | 29.97 | 24.11 | 15.75 | 13.81 | 12.03 | 11.28 | 10.72 | Diverge |
| ZQ-Global* | 28.93/34.29/25.63 | 22.85/27.23/21.50 | 14.80/17.34/14.99 | 12.74/15.32/13.40 | 10.82/13.36/11.91 | 10.23/12.61/11.37 | 9.68/11.95/10.80 | 9.37/11.47/10.38 |
| | 29.62 | 23.86 | 15.71 | 13.82 | 12.03 | 11.41 | 10.81 | 10.41 |

Table E.16: BLOOM W4asym-A16 with various block-size out of the best result from GPTQ and ZQ-Global.

Table E.16: BLOOM W4$^{\text{asym}}$-A16 with various block-size out of the best result from GPTQ and ZQ-Global.

| Method | 560m | 1.1b | 1.7b | 3b | 7.1b | 176b |
|---|---|---|---|---|---|---|
| **BS=1024** | | | | | | |
| RTN | 24.90/46.37/26.68 32.65 | N/A N/A | 16.57/30.14/19.00 21.90 | N/A N/A | 1019.51/1351.45/601.35 990.77 | 53.41/160.05/43.64 85.70 |
| GPTQ | 23.90/43.99/25.47 31.12 | N/A N/A | 16.12/29.13/18.61 21.29 | N/A N/A | 11.57/19.82/14.33 15.24 | 8.16/13.70/11.02 10.96 |
| ZQ-Global | 23.62/43.90/25.41 30.98 | N/A N/A | 15.98/28.67/18.44 21.03 | N/A N/A | 11.91/20.84/14.58 15.78 | 8.23/13.94/11.09 11.09 |
| **BS=512** | | | | | | |
| RTN | 24.78/46.07/26.45 32.44 | 19.41/53.64/21.85 31.63 | 16.47/29.84/18.88 21.73 | 14.29/24.84/17.05 18.73 | 142.38/314.10/100.09 185.52 | 33.88/103.57/31.02 56.16 |
| GPTQ | 23.63/43.96/25.36 30.98 | 18.52/49.73/20.91 29.72 | 16.07/29.87/18.50 21.48 | 13.79/23.77/16.41 17.99 | 11.54/19.75/14.30 15.20 | 8.14/13.70/11.02 10.95 |
| ZQ-Global | 23.50/43.53/25.23 30.75 | 18.31/49.06/20.82 29.40 | 15.93/28.47/18.38 20.93 | 13.82/23.92/16.47 18.07 | 11.85/20.17/14.42 15.48 | 8.20/13.86/11.07 11.04 |
| **BS=256** | | | | | | |
| RTN | 24.09/45.13/26.02 31.75 | 18.87/52.29/21.44 30.87 | 16.27/29.72/18.76 21.58 | 14.16/24.42/16.90 18.49 | 121.09/281.67/88.59 163.78 | 12.55/27.29/15.60 18.48 |
| GPTQ | 23.31/43.43/25.12 30.62 | 18.36/49.13/20.79 29.42 | 16.07/29.10/18.46 21.21 | 13.76/23.61/16.38 17.92 | 11.55/19.72/14.29 15.18 | 8.14/13.70/11.01 10.95 |
| ZQ-Global | 23.17/43.16/25.13 30.49 | 18.24/48.78/20.75 29.26 | 15.81/28.71/18.32 20.95 | 13.79/23.69/16.42 17.97 | 11.59/19.92/14.36 15.29 | 8.17/13.80/11.06 11.01 |
| **BS=128** | | | | | | |
| RTN | 23.82/44.78/25.75 31.45 | 18.62/51.31/21.17 30.37 | 16.13/29.89/18.66 21.56 | 14.00/24.19/16.71 18.30 | 23.90/49.80/24.15 32.62 | 8.84/15.62/11.70 12.06 |
| GPTQ | 23.27/43.10/24.99 30.45 | 18.14/48.72/20.73 29.20 | 16.03/28.96/18.41 21.13 | 13.72/23.65/16.34 17.90 | 11.52/19.73/14.26 15.17 | 8.14/13.67/11.01 10.94 |
| ZQ-Global | 23.14/42.95/24.97 30.35 | 18.17/48.53/20.70 29.13 | 15.75/28.71/18.29 20.92 | 13.73/23.65/16.37 17.92 | 11.56/19.77/14.32 15.22 | 8.17/13.78/11.03 10.99 |
| **BS=64** | | | | | | |
| RTN | 23.65/44.04/25.51 31.07 | 18.53/50.02/21.03 29.86 | 16.06/29.57/18.60 21.41 | 13.93/23.95/16.60 18.12 | 11.85/20.51/14.65 15.67 | 8.31/14.14/11.18 11.21 |
| GPTQ | 23.11/42.95/24.94 30.33 | 18.14/48.87/20.65 29.22 | 16.00/28.91/18.38 21.10 | 13.72/23.68/16.33 17.91 | 11.51/19.70/14.27 15.16 | 8.14/13.69/11.00 10.94 |
| ZQ-Global | 23.00/42.80/24.91 30.24 | 18.10/48.30/20.64 29.01 | 15.68/28.55/18.25 20.82 | 13.70/23.63/16.36 17.90 | 11.53/19.67/14.27 15.16 | 8.17/13.72/11.02 10.97 |
| **BS=32** | | | | | | |
| RTN | 23.60/43.91/25.50 31.00 | 18.63/50.13/21.04 29.93 | 15.98/29.56/18.56 21.37 | 13.92/23.90/16.53 18.12 | 11.65/20.01/14.43 15.36 | 8.20/13.86/11.07 11.04 |
| GPTQ | 23.10/43.19/24.91 30.40 | 18.17/48.35/20.66 29.06 | 15.95/28.95/18.36 21.08 | 13.76/23.60/16.33 17.89 | 11.53/19.71/14.27 15.17 | 8.14/13.70/11.00 10.95 |
| ZQ-Global | 23.07/42.63/24.82 30.18 | 18.07/48.07/20.59 28.91 | 15.66/28.58/18.21 20.82 | 13.72/23.59/16.33 17.88 | 11.52/19.71/14.26 15.16 | 8.16/13.69/11.01 10.95 |

Table E.17: OPT full results of three-bit weight with various block-size.

| Method | 125m | 350m | 1.3b | 2.7b | 6.7b | 13b | 30b | 66b |
|---|---|---|---|---|---|---|---|---|
| **Full Row** | | | | | | | | |
| RTN | 2095.20/1848.83/1222.00 1722.01 | 47.43/53.38/36.93 45.91 | 4399.18/4400.98/3551.88 4117.35 | 8326.78/4208.57/4895.83 5810.40 | 878.00/735.86/910.10 841.32 | 1953.43/1953.60/1669.76 1858.93 | 439.39/691.94/437.96 523.09 | 1465.06/1564.59/1282.58 1437.41 |
| GPTQ | 845.81/599.71/496.14 647.22 | 30.65/34.09/26.15 30.30 | 20.23/27.39/19.45 22.36 | 15.91/19.26/16.01 17.06 | 12.69/15.90/13.96 14.18 | 11.36/13.71/12.21 12.43 | 10.10/12.54/11.20 11.28 | 16.77/21.16/15.39 17.77 |
| ZQ-Global* | 46.47/58.55/35.45 46.82 | 29.64/36.51/25.55 30.57 | 32.48/94.57/28.97 52.01 | 60.91/116.22/36.45 71.19 | 23.87/29.75/23.88 25.83 | 44.70/60.78/46.18 50.55 | 13.16/20.49/13.48 15.71 | 28.93/75.91/27.28 44.04 |
| **BS=1024** | | | | | | | | |
| RTN | N/A N/A | 44.57/49.58/35.09 43.08 | 1950.00/2317.55/1913.55 2060.37 | 3810.79/2563.06/3054.91 3142.92 | 50.01/70.17/99.21 73.13 | 265.62/417.03/261.93 314.86 | 362.47/252.33/364.45 326.42 | 523.81/846.60/1021.17 797.20 |
| GPTQ | N/A N/A | 29.78/33.76/25.66 29.73 | 19.03/23.32/18.14 20.16 | 60.97 N/A | 11.69/14.31/12.70 12.90 | 10.56/12.96/11.70 11.74 | 9.89/12.19/11.02 11.03 | 12.84/16.17/13.02 14.01 |
| ZQ-Global* | N/A N/A | 29.19/34.57/25.11 29.62 | 19.83/29.77/19.79 23.13 | N/A N/A | 13.99/18.82/14.76 15.86 | 13.43/19.28/13.76 15.49 | 11.10/14.46/11.94 12.50 | 11.87/14.86/12.13 12.95 |
| **BS=512** | | | | | | | | |
| RTN | N/A N/A | 37.74/45.10/31.85 38.25 | 1777.53/1304.55/852.03 1311.37 | 1604.07/1407.49/1487.78 1499.78 | 25.13/40.56/40.08 35.26 | 130.75/175.33/135.67 147.25 | 620.53/340.68/416.28 459.16 | 198.01/457.78/426.15 360.65 |
| GPTQ | N/A N/A | 28.46/32.54/25.14 28.71 | 18.02/21.35/17.46 18.94 | 14.38/17.24/14.79 15.47 | 11.57/14.33/12.57 12.82 | 10.41/12.97/11.64 11.67 | 9.77/12.18/10.97 10.97 | 11.89/14.40/12.40 12.92 |
| ZQ-Global* | N/A N/A | 27.81/33.57/24.55 28.65 | 18.31/23.54/17.99 19.95 | 18.10/29.47/17.15 21.57 | 12.54/16.60/13.62 14.25 | 11.82/15.98/12.81 13.54 | 10.48/13.36/11.66 11.83 | 11.26/13.95/11.79 12.33 |
| **BS=256** | | | | | | | | |
| RTN | 4349.14/2907.61/2510.75 3255.84 | 35.36/42.07/30.81 36.08 | 127.17/358.19/142.49 209.28 | 670.51/550.66/531.80 584.32 | 19.10/32.39/27.26 26.25 | 42.52/56.35/43.32 47.40 | 32.84/60.38/33.48 42.23 | 210.01/478.13/413.00 367.05 |
| GPTQ | 41.81/49.95/32.48 41.41 | 27.60/33.73/24.88 28.74 | 16.97/20.19/16.70 17.95 | 13.69/17.06/14.54 15.10 | 11.65/14.24/12.48 12.79 | 10.35/12.93/11.61 11.63 | 9.66/12.10/10.93 10.90 | 11.60/13.98/11.92 12.50 |
| ZQ-Global* | 38.60/46.57/31.36 38.85 | 26.88/32.79/24.08 27.92 | 16.82/21.21/17.05 18.36 | 14.86/19.63/15.37 16.62 | 11.86/15.87/13.10 13.61 | 11.33/14.95/12.48 12.92 | 10.41/12.95/11.41 11.59 | 10.26/12.66/11.08 11.34 |
| **BS=128** | | | | | | | | |
| RTN | 3446.89/2156.26/1484.15 2362.43 | 33.13/41.23/29.51 34.62 | 49.40/88.45/45.07 60.97 | 153.68/155.21/113.98 140.96 | 16.34/26.86/21.98 21.72 | 17.80/25.95/18.28 20.67 | 45.83/43.91/57.50 49.08 | 106.84/241.02/212.94 186.93 |
| GPTQ | 40.00/45.73/31.15 38.96 | 27.66/34.04/25.18 28.97 | 16.47/19.90/16.47 17.61 | 13.81/16.96/14.37 15.05 | 11.57/14.10/12.41 12.69 | 10.35/12.84/11.58 11.59 | 9.73/12.08/10.91 10.91 | 10.96/13.27/11.45 11.90 |
| ZQ-Global* | 36.57/43.88/29.94 36.80 | 25.75/31.59/23.57 26.97 | 16.28/20.20/16.67 17.72 | 14.27/18.41/14.90 15.86 | 11.70/15.05/12.68 13.14 | 11.13/15.07/12.17 12.79 | 10.31/12.99/11.32 11.54 | 10.12/12.66/11.01 11.27 |
| **BS=64** | | | | | | | | |
| RTN | 708.02/477.13/287.03 490.73 | 32.61/42.14/29.09 34.61 | 25.43/38.84/24.63 29.63 | 72.84/69.27/48.07 63.39 | 14.11/21.71/16.56 17.46 | 14.13/20.08/15.25 16.48 | 20.55/32.74/24.49 25.93 | 30.66/70.73/65.57 55.65 |
| GPTQ | 37.15/42.59/30.07 36.60 | 27.68/33.55/25.12 28.78 | 16.25/19.80/16.32 17.46 | 13.66/16.69/14.37 14.91 | 11.42/13.98/12.37 12.59 | 10.37/12.90/11.58 11.62 | 9.68/12.17/10.92 10.92 | 10.39/12.65/11.15 11.40 |
| ZQ-Global* | 35.82/40.98/29.65 35.48 | 25.31/31.60/23.38 26.76 | 16.05/19.77/16.39 17.40 | 13.33/16.92/14.31 14.85 | 11.56/14.70/12.59 12.95 | 10.88/13.64/12.04 12.19 | 10.04/12.70/11.27 11.34 | 10.04/12.06/10.81 10.97 |
| **BS=32** | | | | | | | | |
| RTN | 72.83/88.62/54.25 71.90 | 32.36/40.76/29.06 34.06 | 20.22/27.31/19.81 22.44 | 31.12/42.01/26.83 33.32 | 13.38/18.56/15.44 15.79 | 13.06/18.35/14.38 15.26 | 11.12/15.05/12.35 12.84 | 19.29/43.61/34.10 32.33 |
| GPTQ | 38.26/45.01/30.92 38.06 | 27.16/33.65/24.97 28.59 | 16.13/19.83/16.45 17.47 | 13.66/17.06/14.50 15.07 | 11.43/14.08/12.42 12.64 | 10.48/12.96/11.65 11.70 | 9.78/12.24/10.96 10.99 | Diverge Diverge |
| ZQ-Global* | 33.44/39.48/28.33 33.75 | 25.19/30.73/23.22 26.38 | 15.62/19.52/16.20 17.11 | 13.35/16.64/14.18 14.73 | 11.56/14.38/12.61 12.85 | 10.86/13.64/12.03 12.17 | 10.25/12.86/11.28 11.46 | 9.99/12.05/10.81 10.95 |

Table E.18: BLOOM W3$^{\text{asym}}$-A16 with various block-size out of the best result from GPTQ and ZQ-Global.

| Method | 560m | 1.1b | 1.7b | 3b | 7.1b | 176b |
|---|---|---|---|---|---|---|
| Full row | | | | | | |
| RTN | 68.45/132.83/59.22 86.83 | 118.61/317.41/99.65 178.56 | 31.15/67.23/34.02 44.14 | 31.07/59.03/32.17 40.76 | 66140.72/78568.16/44504.19 63071.02 | 100371.84/166012.19/137892.34 134758.79 |
| GPTQ | 46.92/84.69/39.50 57.04 | 49.78/142.95/43.84 78.85 | 19.70/41.35/21.74 27.59 | 22.84/46.49/22.90 30.74 | 52966.59/52979.88/37115.48 47687.32 | Diverge Diverge |
| ZQ-Global | 33.20/64.61/32.30 43.37 | 34.16/100.05/29.22 54.48 | 19.22/36.30/21.25 25.59 | 18.41/33.10/20.79 24.10 | 273.55/439.59/100.79 271.31 | 27.19/75.74/45.45 49.46 |
| BS=1024 | | | | | | |
| RTN | 47.00/86.57/43.37 58.98 | 70.81/230.74/70.78 124.11 | 35.41/65.75/33.54 44.90 | 22.12/40.65/24.55 29.11 | 25654.77/25531.66/15868.46 22351.63 | 141324.41/183583.73/200436.33 175114.82 |
| GPTQ | 31.25/58.80/30.94 40.33 | N/A N/A | 19.11/37.07/20.90 25.69 | N/A N/A | 12.59/21.95/15.21 16.58 | 8.31/13.96/11.17 11.15 |
| ZQ-Global | 28.91/55.81/29.59 38.10 | N/A N/A | 18.20/34.13/20.40 24.24 | N/A N/A | 30.94/119.98/21.39 57.44 | 15.98/32.85/19.85 22.89 |
| BS=512 | | | | | | |
| RTN | 41.58/79.83/39.41 53.61 | 33.83/116.88/37.34 62.68 | 25.95/49.65/26.77 34.12 | 19.94/38.58/22.58 27.03 | 9777.49/8000.29/5407.46 7728.41 | 202051.34/273707.81/279776.97 251845.38 |
| GPTQ | 28.08/53.15/29.05 36.76 | 21.20/61.42/23.33 35.32 | 18.41/34.47/20.43 24.44 | 15.08/26.14/17.53 19.58 | 12.32/21.29/15.01 16.21 | 8.30/13.98/11.16 11.15 |
| ZQ-Global | 26.80/50.49/28.31 35.20 | 20.77/57.57/22.89 33.75 | 17.64/33.19/19.91 23.58 | 15.16/26.51/17.57 19.75 | 16.35/28.75/15.76 20.29 | 11.38/20.36/14.66 15.47 |
| BS=256 | | | | | | |
| RTN | 36.13/70.37/36.29 47.60 | 28.65/95.72/31.80 52.06 | 21.67/42.59/23.80 29.35 | 17.64/32.82/20.69 23.72 | 1322.61/1864.55/946.92 1378.02 | 166006.80/187829.98/198052.83 183963.20 |
| GPTQ | 27.10/51.11/28.24 35.48 | 20.60/56.57/22.77 33.31 | 17.97/33.28/20.04 23.76 | 14.82/25.79/17.31 19.31 | 12.27/21.24/14.93 16.15 | 8.27/13.99/11.14 11.13 |
| ZQ-Global | 25.96/49.75/27.59 34.43 | 20.21/54.83/22.33 32.46 | 17.43/32.14/19.67 23.08 | 14.85/25.79/17.33 19.32 | 12.85/22.00/15.04 16.63 | 9.07/15.88/11.88 12.28 |
| BS=128 | | | | | | |
| RTN | 34.71/66.56/35.27 45.51 | 24.43/73.77/26.90 41.70 | 19.59/37.22/21.98 26.26 | 16.11/28.81/18.89 21.27 | 108.32/252.15/74.42 144.96 | 111057.84/101926.99/105339.26 106108.03 |
| GPTQ | 26.29/49.86/27.54 34.56 | 20.26/55.76/22.42 32.81 | 17.77/32.65/19.92 23.45 | 14.58/25.25/17.11 18.98 | 12.18/21.06/14.86 16.03 | 8.26/13.92/11.12 11.10 |
| ZQ-Global | 25.28/48.24/26.96 33.49 | 19.79/54.04/22.03 31.95 | 17.12/31.42/19.31 22.62 | 14.62/25.73/17.17 19.17 | 12.04/21.02/14.82 15.96 | 8.43/14.44/11.29 11.39 |
| BS=64 | | | | | | |
| RTN | 30.88/59.01/32.08 40.66 | 23.04/67.93/25.49 38.82 | 19.35/37.67/21.80 26.27 | 15.64/27.56/18.39 20.53 | 37.15/65.22/33.22 45.20 | 198.66/488.11/128.62 271.80 |
| GPTQ | 26.31/49.91/27.17 34.46 | 20.11/55.06/22.23 32.47 | 17.94/32.42/19.76 23.37 | 14.62/25.39/17.07 19.02 | 12.13/21.07/14.83 16.01 | 8.26/13.93/11.11 11.10 |
| ZQ-Global | 25.17/48.01/26.59 33.26 | 19.51/53.27/21.75 31.51 | 16.88/31.14/19.22 22.41 | 14.51/25.18/17.05 18.91 | 12.00/20.85/14.74 15.86 | 8.35/14.06/11.20 11.21 |
| BS=32 | | | | | | |
| RTN | 30.15/57.55/31.51 39.74 | 23.49/70.15/25.56 39.73 | 18.96/36.54/21.42 25.64 | 15.56/27.48/18.32 20.46 | 13.06/23.77/16.05 17.62 | 10.28/18.90/13.27 14.15 |
| GPTQ | 25.96/49.99/27.06 34.33 | 19.97/54.79/22.16 32.31 | 17.60/32.24/19.76 23.20 | 14.55/25.76/17.06 19.12 | 12.20/21.01/14.85 16.02 | 8.28/13.95/11.13 11.12 |
| ZQ-Global | 25.09/47.36/26.34 32.93 | 19.43/52.95/21.64 31.34 | 16.86/30.49/19.11 22.15 | 14.50/25.36/16.99 18.95 | 12.00/20.84/14.72 15.85 | 8.35/14.04/11.20 11.20 |

Table E.19: Full results of BLOOM-176B with different quantization bits

| Bits | 3 | 4 | 5 | 6 | 7 | 8 |
|---|---|---|---|---|---|---|
| Per-row | 27.19/75.74/45.45 | 8.16/13.70/11.02 | 8.13/13.67/10.99 | 8.11/13.63/10.98 | 8.11/13.62/10.97 | 8.10/13.62/10.98 |
| 1024 | 8.31/13.96/11.17 | 8.14/13.70/11.02 | 8.11/13.62/10.97 | 8.11/13.62/10.97 | 8.11/13.63/10.97 | N/A |
| 64 | 8.26/13.93/11.11 | 8.14/13.69/11.00 | 8.11/13.62/10.96 | N/A | N/A | N/A |

Table E.20: OPT full results of Table 5.

| Method | 125m | 350m | 1.3b | 2.7b | 6.7b | 13b | 30b | 66b |
|---|---|---|---|---|---|---|---|---|
| W4$^{\text{asym}}$ full row and A8$^{\text{sym}}$ 128 | | | | | | | | |
| RTN | 36.64/44.84/30.90 37.46 | 25.58/31.06/23.99 26.88 | 19.96/22.31/18.20 20.16 | 18.42/23.01/18.56 20.00 | 12.04/15.92/13.20 13.72 | 10.79/13.65/12.11 12.18 | 10.10/13.17/11.37 11.54 | 20.50/45.58/25.37 30.48 |
| GPTQ | 31.82/38.82/27.54 32.73 | 23.78/28.96/22.61 25.12 | 15.56/18.27/15.62 16.48 | 13.02/15.88/13.76 14.22 | 11.22/13.59/12.11 12.31 | 10.25/12.65/11.37 11.42 | 9.56/11.94/10.79 10.76 | 9.62/11.72/10.54 10.63 |
| ZQ-Local | | | | | | | | 9.79/11.94/10.65 10.79 |
| ZQ-Global | 31.69/36.66/27.19 31.85 | 23.47/28.18/22.03 24.56 | 15.53/18.35/15.73 16.54 | 13.02/16.11/13.82 14.32 | 11.29/13.70/12.19 12.39 | 10.43/12.91/11.64 11.66 | 9.86/12.28/11.00 11.05 | 9.62/11.84/10.63 10.70 |
| W4$^{\text{sym}}$ 128 and A8$^{\text{sym}}$ 128 | | | | | | | | |
| RTN | 30.61/36.57/27.08 31.42 | 24.14/29.47/22.80 25.47 | 15.46/18.68/15.77 16.64 | 13.24/16.36/13.95 14.52 | 11.16/14.08/12.35 12.53 | 10.35/12.89/11.57 11.60 | 9.95/12.15/10.95 11.02 | 9.58/11.90/10.58 10.69 |
| GPTQ | 30.47/36.45/26.45 31.12 | 23.43/28.12/22.06 24.54 | 14.90/17.62/15.17 15.90 | 12.51/15.63/13.48 13.87 | 10.88/13.35/11.93 12.05 | 10.17/12.48/11.28 11.31 | 9.58/11.86/10.74 10.73 | 9.35/11.54/10.40 10.43 |
| ZQ-Local | | | | | | | | 9.40/11.63/10.51 10.51 |
| ZQ-Global | 29.59/34.68/25.91 30.06 | 22.59/27.93/21.68 24.07 | 14.87/17.55/15.11 15.84 | 12.65/15.45/13.48 13.86 | 10.88/13.40/11.94 12.08 | 10.20/12.67/11.43 11.43 | 9.74/12.03/10.83 10.87 | 9.40/11.51/10.42 10.44 |
| W4$^{\text{asym}}$ full row and A8$^{\text{asym}}$ 128 | | | | | | | | |
| RTN | 36.61/44.71/30.85 37.39 | 25.50/30.93/23.88 26.77 | 19.58/22.08/18.01 19.89 | 19.53/24.38/19.68 21.20 | 11.91/15.35/13.01 13.42 | 10.68/13.50/12.02 12.07 | 10.13/13.21/11.37 11.57 | 17.90/32.15/20.02 23.36 |
| GPTQ | 32.15/39.58/27.65 33.13 | 23.48/28.92/22.46 24.95 | 15.43/18.24/15.55 16.40 | 12.92/15.94/13.74 14.20 | 11.17/13.59/12.09 12.29 | 10.35/12.63/11.36 11.45 | 9.65/11.95/10.79 10.80 | 9.58/11.71/10.55 10.61 |
| ZQ-Local | | | | | | | | 10.05/11.91/10.61 10.86 |
| ZQ-Global | 31.55/37.49/27.25 32.10 | 23.34/28.33/22.08 24.58 | 15.52/18.55/15.61 16.56 | 13.07/16.09/13.82 14.33 | 11.32/13.65/12.16 12.37 | 10.42/12.86/11.63 11.64 | 9.86/12.30/11.00 11.05 | 9.67/12.22/10.86 10.91 |
| W4$^{\text{asym}}$ 128 and A8$^{\text{asym}}$ 128 | | | | | | | | |
| RTN | 30.59/36.56/27.07 31.41 | 24.11/29.43/22.74 25.43 | 15.38/18.57/15.69 16.55 | 13.22/16.32/13.91 14.49 | 11.13/13.97/12.30 12.47 | 10.34/12.82/11.55 11.57 | 9.98/12.15/10.96 11.03 | 9.57/11.86/10.58 10.67 |
| GPTQ | 30.47/36.19/26.40 31.02 | 23.35/27.96/21.94 24.42 | 14.92/17.57/15.12 15.87 | 12.48/15.60/13.46 13.85 | 10.87/13.34/11.91 12.04 | 10.20/12.45/11.28 11.31 | 9.62/11.88/10.74 10.75 | 9.39/11.55/10.41 10.45 |
| ZQ-Local | | | | | | | | 9.37/11.70/10.49 10.52 |
| ZQ-Global | 29.85/34.52/26.10 30.16 | 22.70/27.72/21.64 24.02 | 14.96/17.55/15.09 15.86 | 12.64/15.40/13.47 13.84 | 10.93/13.43/11.95 12.10 | 10.18/12.68/11.42 11.42 | 9.74/12.02/10.83 10.86 | 9.39/11.53/10.42 10.45 |

Table E.21: BLOOM full results of Table 6.

| Method | 560m | 1.1b | 1.7b | 3b | 7.1b | 176b |
|---|---|---|---|---|---|---|
| **W4$^{asym}$ full row and A8$^{sym}$ 128** | | | | | | |
| RTN | 25.32/46.98/27.12 33.14 | 23.87/68.29/25.97 39.38 | 16.99/31.15/19.51 22.55 | 14.69/25.22/17.30 19.07 | 12.07/20.86/14.84 15.92 | 8.34/14.05/11.24 11.21 |
| GPTQ | 24.00/44.47/25.66 31.37 | 24.14/66.95/26.17 39.09 | 16.38/29.64/18.79 21.61 | 14.10/24.19/16.67 18.32 | 11.77/20.22/14.48 15.49 | 8.20/13.82/11.07 11.03 |
| ZQ-Local | | | | | | 8.30/14.01/11.20 11.17 |
| ZQ-Global | 23.92/44.23/25.69 31.28 | 22.53/57.71/23.51 34.58 | 16.25/29.72/18.74 21.57 | 14.12/24.26/16.74 18.38 | 11.78/20.30/14.53 15.53 | 8.24/13.82/11.10 11.05 |
| **W4$^{asym}$ 128 and A8$^{sym}$ 128** | | | | | | |
| RTN | 23.84/44.94/25.79 31.53 | 18.65/51.54/21.21 30.46 | 16.18/30.03/18.70 21.64 | 14.04/24.32/16.77 18.38 | 23.05/48.33/23.69 31.69 | 8.87/15.68/11.72 12.09 |
| GPTQ | 23.22/43.24/25.01 30.49 | 18.25/48.89/20.74 29.29 | 16.00/29.44/18.41 21.29 | 13.77/23.68/16.35 17.93 | 11.54/19.76/14.27 15.19 | 8.13/13.69/11.01 10.95 |
| ZQ-Local | | | | | | 8.20/13.87/11.08 11.05 |
| ZQ-Global | 23.12/43.22/25.03 30.45 | 18.19/48.96/20.72 29.29 | 15.75/28.81/18.30 20.95 | 13.73/23.65/16.39 17.92 | 11.57/19.85/14.32 15.25 | 8.17/13.76/11.03 10.99 |
| **W4$^{asym}$ full row and A8$^{asym}$ 128** | | | | | | |
| RTN | 25.30/46.87/27.10 33.09 | 23.90/68.31/25.98 39.39 | 16.96/31.09/19.48 22.51 | 14.68/25.19/17.28 19.05 | 12.07/20.86/14.84 15.92 | 8.34/14.06/11.24 11.21 |
| GPTQ | 23.97/44.15/25.62 31.24 | 24.61/68.19/26.53 39.78 | 16.36/29.77/18.81 21.65 | 14.10/24.17/16.66 18.31 | 11.78/20.32/14.49 15.53 | 8.20/13.82/11.07 11.03 |
| ZQ-Local | | | | | | 8.32/13.97/11.20 11.16 |
| ZQ-Global | 23.88/44.40/25.68 31.32 | 22.63/57.91/23.39 34.64 | 16.25/29.77/18.74 21.59 | 14.17/24.24/16.74 18.38 | 11.77/20.28/14.52 15.52 | 8.25/13.82/11.10 11.06 |
| **W4$^{asym}$ 128 and A8$^{asym}$ 128** | | | | | | |
| RTN | 23.83/44.89/25.77 31.50 | 18.63/51.46/21.19 30.43 | 16.16/29.95/18.68 21.60 | 14.03/24.27/16.75 18.35 | 23.51/49.07/23.96 32.18 | 8.85/15.65/11.72 12.08 |
| GPTQ | 23.26/43.24/25.00 30.50 | 18.18/48.84/20.73 29.25 | 16.05/29.34/18.42 21.27 | 13.69/23.56/16.34 17.86 | 11.54/19.75/14.28 15.19 | 8.14/13.71/11.02 10.96 |
| ZQ-Local | | | | | | 8.19/13.90/11.07 11.06 |
| ZQ-Global | 23.12/43.14/25.01 30.42 | 18.18/48.99/20.73 29.30 | 15.71/28.73/18.30 20.91 | 13.74/23.68/16.39 17.94 | 11.56/19.85/14.31 15.24 | 8.17/13.78/11.04 11.00 |

Table E.22: Full results of Table 6.

| Block SIze | 1024 | 512 | 256 | 128 | 64 | 32 |
|---|---|---|---|---|---|---|
| PPL | 8.16/13.75/11.04 | 8.15/13.75/11.02 | 8.15/13.70/11.01 | 8.13/13.69/11.01 | 8.14/13.69/11.01 | 8.14/13.69/11.01 |

Table E.23: Results of applying LoRC on top of ZQ-Global for INT8 Activation.

| model-size | precision | LoRC-dim | Learning Rate | | | | | |
|---|---|---|---|---|---|---|---|---|
| | | | 0.0005 | 0.0001 | 5.00E-05 | 1.00E-05 | 5.00E-06 | Best |
| 125m | W4A8 | 0 | 4482.1 | 31.15 | 30.40 | 30.55 | 30.72 | 30.40 |
| | | 8 | 5996.14 | 30.96 | 30.24 | 30.37 | 30.61 | 30.24 |
| | | 16 | 3577.12 | 31.02 | 30.26 | 30.2 | 30.37 | 30.20 |
| 125m | W3A8 | 0 | 4283.28 | 41.03 | 40.93 | 55.74 | 86.34 | 40.93 |
| | | 8 | 2396.92 | 37.25 | 36.65 | 37.85 | 39.06 | 36.65 |
| | | 16 | 1787.74 | 36.66 | 36.55 | 37.46 | 38.21 | 36.55 |
| 125m | W2A8 | 0 | 3473.18 | 583.72 | 996.76 | 2480.69 | 3203.11 | 583.72 |
| | | 8 | 3815.37 | 144.85 | 160.71 | 362.17 | 466.98 | 144.85 |
| | | 16 | 3324.23 | 135.25 | 156.28 | 295.78 | 372.7 | 135.25 |
| | | | Learning Rate | | | | | |
| | | LoRC-dim | 5.00E-05 | 1.00E-05 | 5.00E-06 | 1.00E-06 | 5.00E-07 | best |
| 350m | W4A8 | 0 | 25.65 | 24.38 | 24.34 | 24.55 | 24.75 | 24.34 |
| | | 8 | 25.56 | 24.3 | 24.24 | 24.45 | 24.66 | 24.24 |
| | | 16 | 25.45 | 24.39 | 24.21 | 24.39 | 24.63 | 24.21 |
| 350m | W3A8 | 0 | 30.59 | 28.45 | 28.94 | 31.51 | 32.39 | 28.45 |
| | | 8 | 30.1 | 28.22 | 28.71 | 30.81 | 32.09 | 28.22 |
| | | 16 | 30.64 | 28.02 | 28.50 | 30.62 | 31.69 | 28.02 |
| 350m | W2A8 | 0 | 97.40 | 177.43 | 257.61 | 668.19 | 722.19 | 97.4 |
| | | 8 | 95.79 | 139.68 | 194.36 | 437.18 | 459.92 | 95.79 |
| | | 16 | 106.51 | 137.81 | 172.93 | 400.91 | 421.59 | 106.51 |