# OpenReview forum: "From Comprehensive Study to Low-Rank Compensation: Exploring Post-Training Quantization in LLMs"
_NeurIPS.cc/2023/Conference — Submitted to NeurIPS 2023_

### Official Review · Reviewer_x1uC · 2023-07-01

**Soundness:** 3 good
**Presentation:** 3 good
**Contribution:** 3 good
**Rating:** 7
**Confidence:** 4

**Summary:**

This paper conducts a comprehensive analysis of various quantization methods for large language models (LLMs). Some interesting takeaways were shared, for example, activation quantization is generally more susceptible to weight quantization; none of the current quantization methods can achieve the original model quality. Based on such insights, the paper also proposes an optimized method called Low-Rank Compensation (LoRC), which employs low-rank matrices to enhance model quality recovery with a minimal increase in model size.

**Strengths:**

- The motivation of the paper is solid. With the rapid development of LLM, it is essential to study the methodology to deploy LLM on more accessible hardware, where quantization is an important category of the approach. Therefore, a comprehensive study of these methods is necessary.

- The insights shared by this paper are helpful, e.g., it is interesting to know activation quantization is generally more susceptible to weight quantization.


- The proposed improvement based on the observation is reasonable.



**Weaknesses:**

- The proposed method is based on low-rank approximation, which can be viewed as a sparsification-based method. It is kind of out of the scope of the proposed method.


**Questions:**

- Is it possible to open-source the code for reproducibility?


**Limitations:**

Not applicable.

---

> ### Author Rebuttal · Authors · 2023-08-09
>
> Thank you for valuing our research's importance in LLM deployment and acknowledging the relevance of our insights on quantization. We're pleased our proposed improvements resonated and welcome further feedback.
>
> *Q1:* The proposed method is based on low-rank approximation, which can be viewed as a sparsification-based method. It is kind of out of the scope of the proposed method.
>
> *A1:*  Thank you for pointing out the connection between our method and sparsification through low-rank approximation.  Our primary intention was to leverage the benefits of such approximations to enhance the quality of quantization in LLMs.  We believe that utilizing tools from related compression domains can provide innovative solutions, even if they originate from seemingly different approaches. Other quantization work also utilizes various methods to reduce the quantization error, e.g.,[1] uses both int8 and fp16 to represent a single weight matrix.
>
> [1] Dettmers, Tim, et al. "Llm. int8 (): 8-bit matrix multiplication for transformers at scale." arXiv preprint arXiv:2208.07339 (2022).
>
> -----
>
> *Q2:* Is it possible to open-source the code for reproducibility?
>
> *A2:* Yes, we will release the codes.

---

### Official Review · Reviewer_iFFk · 2023-07-01

**Soundness:** 3 good
**Presentation:** 3 good
**Contribution:** 3 good
**Rating:** 5
**Confidence:** 2

**Summary:**

This paper analyzes post-training quantization (PTQ) techniques in large language models, exploring various schemes, model families, and bit precision. The authors propose an optimized method called Low-Rank Compensation (LoRC) to enhance model quality recovery with minimal size increase.

**Strengths:**

1. An evaluation and comparison of existing PTQ methods provide some insights for the community.
2. The paper is well-written.

**Weaknesses:**

Reading from paper provides a satisfying experience, but I must admit that my understanding of the LLM field is limited. Thus, my suggestions may be wrong, please directly point them out.

1. The points in Figure 1 are too dense and lack recognition.
2. Although the method is proposed for LLMs, it's also better to compare it with some traditional quantization methods on ResNet-series.

**Questions:**

The authors could refer to the Weaknesses.

---

> ### Author Rebuttal · Authors · 2023-08-09
>
> Thanks for finding our paper easy to read and understand. We appreciate your positive feedback.
>
> ----
> *Q1:*  The points in Figure 1 are too dense and lack recognition.
>
> *A1:*   Thanks for the suggestions. We thanks reviewer pointing out that figure 1 is hard to differentiate the difference of different methods. We will do the following for the final revision
> - (1) adding higher resolution images in the appendix for better visual quality;
> - (2) adding tables for several bin sizes for easy reading.
>
> ----
>
> *Q2:*  Although the method is proposed for LLMs, it's also better to compare it with some traditional quantization methods on ResNet-series.
>
> *A2:*   Thank you for the suggestion to compare our method with traditional quantization techniques on the ResNet-series. We recognize the value of such a comparison in demonstrating the versatility and breadth of our method. However,
>  - (1) Applying ResNet-series method to LLMs might be challenge since most of them rely on heavy computer cost and time consuming. Please see the literature Reviewer wnse mentioned.
>  -  (2) Our primary focus in this study has been on Large Language Models (LLMs) due to their unique characteristics and challenges in quantization (heavy compute cost and time consuming). Integrating a comparison with ResNet-series on computer vision tasks would require extensive evaluation in the domain of computer vision, which is beyond the scope of our current work.
> - (3) Also, the key component LoRC is used to maintain good accuracy while introducing minimal the memory footprint. CNNs are usually more compute heavy but the model size is relatively small.
>
> LoRC can be used as a simple add-on component for different quantization methods. As such, we applied this to ViT-large on ImageNet (google/vit-large-patch16-224  from Hugging Face) using PTQ (particularly, we here use RTN). Table 1 shows the Top-1 accuracy results of per-row weight quantization with or without LoRC (rank=8). As can be seen, there is a significant accuracy boost from LoRC. Particularly, W2A16 using LoRC can achieve even better accuracy than W3A16 using RTN.
>
> Table 1: The accuracy improvement from LoRC for ViT-Large model.
> |       Using LoRC   |      W16A16 (baseline)  |      W4A16  |      W3A16  |      W2A16  |
> |--------------------|-------------------------|-------------|-------------|-------------|
> |      No            |     82.878              |     82.642  |     68.858  |     0.126   |
> |      Yes (rank 8)  |     N/A                 |     82.754  |     81.902  |     73.480  |

---

> > ### Comment · Reviewer_iFFk · 2023-08-17
> >
> > Thank the authors for the explanation, and all my concerns are addressed. Since I am not an expert in this field, so I finally decided to keep my score.

---

### Official Review · Reviewer_LU91 · 2023-07-04

**Soundness:** 2 fair
**Presentation:** 2 fair
**Contribution:** 2 fair
**Rating:** 4
**Confidence:** 3

**Summary:**

This paper studied the post-training quantization method for 4-bit weight quantization and W4A4 quantization. The authors further proposed a Low-Rank Compensation (LoRC), to enhance model quality with low-rank matrices.

**Strengths:**

The paper is well-written and easy to follow.

**Weaknesses:**

The novelty is not very significant. The post-training quantization with finer-grained and zero-shift is not a new idea.

The sensitivity analysis is conducted with one particular quantization method, and the conclusion should be conditioned on that quantization method, but not general enough to conclude that PTQ exhibits the same behavior. I would suggest the authors to compare different quantization functions, such as minmax/percentile to deliver a more comprehensive conclusion.

The accuracy improvement is marginal. Figure 1 didn’t show much improvement compared to the previous naïve baseline of RTN.


**Questions:**

Have the authors tried other different baseline quantization method, e.g., minmax/percentile quantization?

---

> ### Author Rebuttal · Authors · 2023-08-10
>
> We thank the reviewer for your comments and appreciate the opportunity to address your concerns. As we strive to maintain the highest level of integrity in our research, we welcome any feedback that will help us improve our work.
>
> ---
>
> *Q1:*  The novelty is not very significant. The PTQ with finer-grained and zero-shift is not a new idea.
>
> *A1:*  Sorry for the confusion. We did not claim fine-grained quantization and zero-shift as the contribution of our paper. Those help us to conduct comprehensive study on LLM quantization. The novel part of our work is described in Section 5, which provides a simple but effective way to further boost the quantization performance. Also, we want to emphasize the comprehensive study also provides non-trivial value to the community for further exploration.
>
> ---
> *Q2:*  The conclusion should be conditioned on that quantization method, but not general enough to conclude that PTQ exhibits the same behavior. I would suggest the authors compare different quantization functions, such as minmax/percentile to deliver a more comprehensive conclusion.
>
> *A2:*  Thanks a lot for the suggestions. We are not sure about what minmax quantization is different than the baseline RTN used in the paper (RTN is using the min/max to perform the quantization). Please let us know if we misunderstand minmax quantization.
>
>  - For percentile quantization, in computer vision, it normally outperforms RTN on outlier quantization (particularly for activation quantization), which has been shown in HAWQ series of work in their Github repository ([1-3]) (use RTN for weight quantization and optional percentile for activation quantization). However, For LLM this has been shown to be oppoosite. We will give more details backed up by our additional experiments.
>
> - Also, we want to emphasize that fine-grained (either block-wise or per-row/token wise) is also an effective way to reduce the outlier-effect of activation quantization as demonstrated in Table 5 for activation quantization.
>
> - Finally, we want to note is that percentile quantization for activation is usually not used together with dynamic token-wise quantization as the initial goal of percentile quantization is to reduce the dynamic/outlier-effect of the activation. To use both percentile and dynamic (per-token or block wise) will significantly increase the on-the-fly activation quantization cost.
>
> [1] Dong, et al. "Hawq: Hessian aware quantization of neural networks with mixed-precision." Proceedings of the IEEE/CVF International Conference on Computer Vision. 2019.
>
> [2] Dong, et al. "Hawq-v2: Hessian aware trace-weighted quantization of neural networks." Advances in neural information processing systems 33 (2020): 18518-18529.
>
> [3] Yao et al. "Hawq-v3: Dyadic neural network quantization." International Conference on Machine Learning. PMLR, 2021.
>
>
> Additional experiments:
> As can be seen in the following table (Table 1), the weight only percentile quantization does not provide better accuracy than standard RTN quantization. For activation only and weight-and-activation percentile quantization, the accuracy drop of percentile is much larger than standard RTN. We have spent some time to understand this phenomenon and find the outliers of activation plays a crucial role to preserve the accuracy of LLMs. Similar finding can be found in SmoothQuant work (2211.10438, arxiv.org). In Table 5, SmoothQuant compares their results with Outlier Suppression (2209.13325, arxiv.org), one of the main differences between those two algorithms is that besides both migrated activation quantization difficulty to weight, Outlier Suppression also clipped the activation range (similar to percentile quantization), and this leads to significant accuracy degradation.
>
> Table 1: Compare between percentile (0.1% and 99.9%) and min/max. For W4A8, quantization percentile applied to both weight and activation.  For W16A8 (W4A16), quantization percentile applied to activation (weight). All the quantization results below use the configuration: row-wise weight RTN quantization and token-wise activation RTN quantization.
> | precision  | quantization percentile | 1.3b    | 6.7b    | 13b     | 65b     |
> |-------|-------|---------|---------|---------|---------|
> | W4A16      | no                      | 19.77   | 13.44   | 12.09   | 11.52   |
> | W4A16      | yes                     | 23.27   | 14.58   | 13.96   | 11.74   |
>
> | precision  | quantization percentile | 1.3b    | 6.7b    | 13b     | 65b     |
> |------------|-------------------------|---------|---------|---------|---------|
> | W4A8       | no                      | 21.21   | 14.81   | 26.34   | 84.41   |
> | W4A8       | yes                     | 7958.96 | 7452.49 | 7649.71 | 5338.85 |
>
> | precision  | quantization percentile | 1.3b    | 6.7b    | 13b     | 65b     |
> |-------|------------|---------|---------|---------|---------|
> | w16A8 | no         | 15.99   | 12.55   | 15.38   | 23.74   |
> | w16A8 | yes        | 5002.91 | 8100.55 | 7619.81 | 5539.92 |
>
>
>
> ---
>
> *Q3:* The accuracy improvement is marginal. Figure 1 didn’t show much improvement compared to the previous naïve baseline of RTN.
>
> *A3:* Sorry for the confusion, due to the image size, it is hard to differentiate the difference between different methods. A more clear conclusion can be found in Table 2/Table 4 (and its full versions in Table E.15/E.16) for RTN and Table 7 for LoRC. For OPT-6.7B (per-row/block-size 256) quantization, RTN’s PPL is 13.44/12.57 and LoRC’s PPL is 12.10/11.99. The improvements are 1.34 and 0.58 respectively. For OPT-66B (per-row/block-size 256) quantization, RTN’s PPL is 31.52/10.80 and LoRC’s PPL is 10.34/10.29. Note that the full precision OPT-66B's PPL is 10.33. LoRC almost achieves no degradation. We will add a small discussion to compare our method and RTN in the final revision for easier across comparison.
>
> We also take action to improve the figures and the way we present our results. See our A1 to Q1 from Reviewer **iFFk**.

---

### Official Review · Reviewer_wnse · 2023-07-06

**Soundness:** 3 good
**Presentation:** 3 good
**Contribution:** 3 good
**Rating:** 6
**Confidence:** 5

**Summary:**

This work focuses on systematic examination of various post training quantization techniques in large language models. Experimental analysis include comparison of different model sizes, different numerical precision, and quantization of only weights vs activations. In addition, the Low Rank Compensation method is proposed to enhance model quality recovery.

**Strengths:**

The research on LLMs is rapidly growing, but computational and memory capabilities are still limited when deploying huge models. Therefore, quantization is a must. The analysis presented in this work is definitely needed to advance LLMs deployment in multiple use cases. The paper flows logically and is well structured. The proposed compensation method is interesting.

**Weaknesses:**

1. Quantization may be input data specific. Same model topology when trained on different data may behave differently. It’s mentioned that you “use the zero-shot validation perplexity (PPL) differential on three datasets, namely, Wikitext-2 [23], PTB [22], and C4 [27], before and after the quantization”, however results presented in the following tables doesn’t indicate what error was achieved on which dataset. Could you clairfy?
2. Quantization may be operator specific. It would be interesting to list operators present in both model topologies and identify layers that were specifically sensitive to quantization.

**Questions:**

1. Please clarify the finding about the layer norm in the OPT family. Were all weights and biases 1 and 0 in all layer norm layers? Are you sure the model was initiated correctly?
2. Does you compensation technique require any fine tuning? It wasn’t clear from the text.
3. How does your compensation technique compare to e.g., LAPQ https://arxiv.org/pdf/1911.07190.pdf where quantization error is used for layer smoothing in a similar way.
4. Does your compensation technique work in a layer-by-layer manner? How can you ensure that error doesn’t accumulate across layers?

**Limitations:**

Listed limitations and future work directions are clear and make sense.

---

> ### Author Rebuttal · Authors · 2023-08-09
>
> We are glad that you find our work significant & timely in advancing the deployment of LLMs. We appreciate your positive feedback on the logical flow & structure of our paper, and on our proposed compensation method. Please find our responses below:
>
> *Q1:* Results in tables are not clear
>
> *A1:* The results presented in the main text are the average of the three datasets. For detailed results, please refer the appendix, e.g., the corresponding detailed results of Table 2 is given in Table E.1 as described in the caption of Table 1.
>
> ---
>
> *Q2:* list operators & layers that were sensitive to quant.
>
> *A2:* Thank you for the suggestions. We conduct experiments on OPT-1.3B & 6.7B. The structure of our tests was as follows: (i) sensitivity of four components (QKV, Attn-out, MLP1, MLP2) and the results are in Table 1 (ii) sensitivity of three-type layers (first/middle/final 1/3 layers) and the results are in Table 2.
>
> Table 1. Results for W4A8 and W4A16 using RTN across various linear modules. Observations suggest that the QKV component has the highest sensitivity during quantization. Following QKV, MLP1 exhibits significant sensitivity, while Attn-out appears to be the least sensitive. Baseline for 1.3b is 15.44 PPL, and for 6.7b is 11.89 PPL. Note the values are the mean of three datasets c3, ptb & wikitext-2, consistent in the tables of our submission.
>
> | size | bits | QKV   | ATT-out | MLP1   | MLP2   |
> |------|-------|-------|---------|-------|-------|
> | 1.3b | w4a16 | 16.25 | 15.52   | 16.24 | 15.82 |
> | 1.3b | w4a8 | 17.04 | 15.52   | 16.29 | 15.82 |
> | 6.7b | w4a16 | 12.29 | 11.92   | 12.26 | 12.01 |
> | 6.7b | w4a8 | 13.28 | 11.92   | 13.15 | 12.01 |
>
> Table 2. Outcomes of W4A8 and W4A16 using RTN on different layer divisions. The data indicates pronounced sensitivity to quantization in the initial one-third of the layers, showing the most adverse perplexity. Conversely, the mid and final thirds don't follow a definitive pattern.
>
> | size | bits | first 1/3 | middle 1/3 | last 1/3 |
> |------|-------|------|------|------|
> | 1.3b | w4a16 | 17.21 | 16.07 | 16.11 |
> | 1.3b | w4a8 | 17.52 | 16.10 | 17.54 |
> | 6.7b | w4a16 | 12.46 | 12.31 | 12.12 |
> | 6.7b | w4a8 | 12.80 | 12.52 | 12.17 |
>
> The above observed potential of mixed-precision quantization across different layers and components suggests a tailored approach might be more effective. Instead of a uniform quantization method, a customized strategy for each layer, based on its sensitivity, could optimize computational efficiency without major accuracy trade-offs. The additional experiments suggested by the reviewer pave the way for future research on formulating dynamic quantization algorithms and understanding the reasons behind these sensitivities.
>
> ---
>
> *Q3:* Clarify the LN in OPT
>
> *A3:* We double checked our evaluation and the initiation is correct. We further checked if the LN of opt-family are well trained or not using the following code snippet. See below for the code snippets (due to space limit, we only show one LN but rest are similar). We will release codes for reproducibility. We mistakenly said the bias is not well-trained for models and will correct this in the final revision.
>
> The code snippets for OPT:
> ```
> from transformers import OPTForCausalLM
> model = OPTForCausalLM.from_pretrained('facebook/opt-6.7b')
> for n, p in model.named_parameters():
>       if 'layer_norm' in n:
>           print (n, "end=:")
>           if "weight" in n:
>                print(((p-1).abs() > 1e-4).sum())
>           else:
>                print((p.abs() > 1e-4).sum())
> ```
> Our results  for 6.7b model:
> ```
> ....
> model.decoder.final_layer_norm.weight end=: tensor(0)
> model.decoder.final_layer_norm.bias end=: tensor(4094)
> ```
> for 66b model:
> ```
> ...
> model.decoder.final_layer_norm.weight end=: tensor(0)
> model.decoder.final_layer_norm.bias end=: tensor(9200)
> ```
>
> ---
>
> *Q4:* LoRC requires any finetuning?
>
> *A4:* Thanks for the question, our method LoRC works actually for both post-training quantization and quantize aware training. Due to the huge amount of resource and time requirements, this paper focuses only on post-training quantization exploration as indicated in the title of our manuscript. Thus, the results shown in Table 7 require no fine-tuning (see line 290-296 and footnote 6).
>
> ---
>
> *Q5:* Compare LoRC to e.g., LAPQ.
>
> *A5:* Thanks for pointing out this work, and we will include it in related work in our final revision. The LAPQ work focuses on the different quantization step size and how to jointly optimize it using Hessian and iterative Powell optimization. However, our work LoRC is more about after quantization, how we can further boost the accuracy by incorporating the error matrix with a low rank decomposition (see A2 of Reviewer iFFk). The two methods should be easily compossible.
>
> Other than above, LAPQ methods might be too expensive (both compute-/time-wise): (1) it needs to compute the Hessian across different layers. The LLM has much larger parameter space and more layers as compared to ResNet & MobileNet. (2) Similarly, the iterative optimization is also not cheap. Those drawbacks may affect the application of LAPQ on LLMs.
>
> ---
>
> *Q6:* Error accumulates across layers.
>
> *A6:* This is a great point. Yes, our method is working in a layer-by-layer manner. There is no guarantee about the error accumulation across different layers. However, this is a common issue of LLMs quantization, e.g., ZeroQuant [36], SmoothQuant [35] and LLM.int8 [6]. All works are not able to deal with across-layer error accumulation and one of the core reasons is about the complexity and cost of LLMs. Also, we note that although this is not taken into current algorithmic considerations, the overall quantization performance of LLMs is good for most of cases. However, this is an interesting direction to explore, particularly for extreme low-precision quantization, e.g., 2/3 bits. We will include this as a further opportunity in our discussion.

---

> > ### Comment · Reviewer_wnse · 2023-08-14
> > **Response to rebuttal**
> >
> > Thank you for clarification and a very detailed explanation. All my questions were addressed.

---

### Decision · Program_Chairs · 2023-09-21

**Decision:**

Reject

**Comment:**

This paper provides systematic studies on the existing post-training quantization methods, and also proposed a low-rank compensation method to improve the performance. The main concern is around the novelty and technical contribution, while the presentation can also be improved in terms of the clarity. With that said, we do believe that the paper has made nontrivial contribution to the field, and we highly encourage the authors to refine the paper further and we would envision a high quality revision next.